# 🍎 APPLE: Toward General Active Perception via Reinforcement Learning

**Tim Schneider**[1,2]**, Cristiana de Farias**[1]**, Roberto Calandra**[3]**, Liming Chen**[2]**, and Jan Peters**[4]

[1]*Department of Computer Science, TU Darmstadt, Germany.*
[2]*LIRIS, CNRS UMR5205, École Centrale de Lyon, France.*
[3]*LASR Lab & CeTI, TU Dresden, Germany.*
[4]*DFKI, Hessian.AI, RIG, and Centre for Cognitive Science, TU Darmstadt, Germany.*
 Corresponding author: `tim@robot-learning.de`

## Abstract

Active perception is a fundamental skill that enables us humans to deal with uncertainty in our inherently partially observable environment. For senses such as touch, where the information is sparse and local, active perception becomes crucial. In recent years, active perception has emerged as an important research domain in robotics. However, current methods are often bound to specific tasks or make strong assumptions, which limit their generality. To address this gap, this work introduces `APPLE` (**A**ctive **P**erception **P**olicy **Le**arning) – a novel framework that leverages reinforcement learning (RL) to address a range of different active perception problems. `APPLE` jointly trains a transformer-based perception module and decision-making policy with a unified optimization objective, learning how to actively gather information. By design, `APPLE` is not limited to a specific task and can, in principle, be applied to a wide range of active perception problems. We evaluate two variants of `APPLE` across different tasks, including tactile exploration problems from the Tactile MNIST benchmark. Experiments demonstrate the efficacy of `APPLE`, achieving high accuracies on both regression and classification tasks. These findings underscore the potential of `APPLE` as a versatile and general framework for advancing active perception in robotics.

**Project page:** https://timschneider42.github.io/apple

## 1 Introduction

Imagine searching for a set of tools inside a cluttered toolbox. You do not know where a tool is located or how it is positioned. Rather than waiting passively for the information to reveal itself, most people would place their hands inside the box and begin exploring. They would probe and adjust their motions based on the feedback received. This process illustrates the concept of *active perception*: the deliberate selection of actions to acquire information in the face of uncertainty Bajcsy (1988). Active perception does not aim to exhaustively explore every aspect of the world; it focuses on finding efficient strategies that reduce uncertainty about specific properties of the environment. Equipping robots with this capability is a key step toward enabling them to act autonomously in unstructured environments, where information is sparse, noisy, and incomplete.

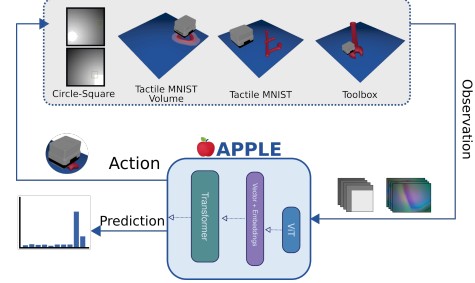

Figure 1: Our method *Active Perception Policy Learning* (`APPLE`) aims to infer properties, such as object classes, of its environment based on limited per-step information. To do so, it jointly optimizes an action policy for information gathering and a prediction model for inference. Both the action policy and prediction models use a shared transformer-based backbone to process input sequences. Shown at the top are four benchmark tasks we use to evaluate `APPLE`.

Particularly relevant to active perception is the sense of touch. While vision has been widely explored in this context, especially for tasks such as object search and next-best-view planning Yang et al. (2019); Xiong et al. (2025a); Tan et al. (2020), tactile sensing poses distinct challenges. Unlike vision, which can provide wider information coverage from a single observation, touch is inherently local, with each contact providing only a small glimpse of the environment Prescott et al. (2011). This locality makes tactile sensing a natural fit for active perception, since purposeful interaction is often the only practical way to gather meaningful information. Several works have investigated active tactile perception, e.g., for shape estimation Björkman et al. (2013); Smith et al. (2021), texture recognition Boehm et al. (2024), and grasping De Farias et al. (2021). However, these approaches are typically algorithmically tailored to a task-specific objective, such as maximizing force closure for grasping or reconstructing shape, often using greedy information-gain heuristics. Moreover, methods such as Björkman et al. (2013); Dragiev et al. (2013); De Farias et al. (2021) simplify the problem by assuming that objects remain stationary during exploration, an assumption that overlooks the dynamic and contact-rich nature of tactile exploration. While effective in narrow settings, such methods remain fundamentally tied to predefined objectives and assumptions. In contrast, alternative options, such as reinforcement learning (RL), can provide a more general framework for acquiring more dynamic active perception policies. RL enables agents to *learn sequential decision-making strategies* directly from interaction, guided by broader task-level goals rather than hand-crafted optimization criteria.

Some works have explored RL for learning active perception strategies. For tactile sensing, various RL algorithms have been employed, including REINFORCE Fleer et al. (2020), PPO Xu et al. (2022); Shahidzadeh et al. (2024), and DQN Smith et al. (2021). Notably, Fleer et al. (2020) introduced the Haptic Attention Model (`HAM`), which adapts the recurrent model of visual attention proposed by Mnih et al. (2014) to the tactile domain. `HAM` demonstrated that even simple policy gradient methods such as REINFORCE can learn exploration strategies by jointly optimizing perception and action, enabling haptic object classification. However, on-policy approaches, such as REINFORCE and PPO, can be sample-inefficient, limiting their scalability. Moreover, the generality of active perception methods has not been widely studied: most existing approaches are tied to specific tasks and objectives, lacking a unified formulation. Ideally, the formulation of such methods should be agnostic to the underlying task, enabling transfer across modalities and problem settings.

Thus, in this work, we ask the following: can we design a principled RL-based algorithm for discovering active perception policies by relying only on a ground-truth label and a differentiable loss during training? And more broadly, can such an approach be general enough to extend across a diverse set of active perception problems, ranging from classification to regression, without requiring task-specific exploration heuristics? To investigate these questions, we frame active perception within the setting of partially observable Markov decision processes (POMDPs), where an agent must act under uncertainty to actively reduce ambiguity about a target property. Note that in this work, we purely aim to evaluate the agent's ability to actively perceive. Accordingly, we focus on tasks where the agent's primary objective is to learn a property of the environment (e.g., the class or pose of an object) and leave the evaluation of problems that involve active perception for another downstream task to future work. Building on this formulation, we introduce `APPLE` (**A**ctive **P**erception **P**olicy **Le**arning), a framework that combines reinforcement learning with supervised learning, requiring only a differentiable loss function and a POMDP environment. `APPLE` jointly trains a decision-making policy and a perception module on top of a shared transformer backbone, allowing it to accommodate diverse sensor inputs without task-specific modifications. We present two variants of `APPLE`, extending `SAC` Haarnoja et al. (2018) and `CrossQ` Bhatt et al. (2019), and evaluate them across five benchmarks that include classification, volume estimation, and localization tasks (see Fig. 1). Our experiments show that our RL framework can be effective for active perception in several tasks and serves as a step towards a more general framework. Our main contributions are:

- A unified formulation for active perception that motivates using a combination of policy gradient methods and supervised learning to solve interactive supervised learning problems.

- A framework for active perception that jointly trains a reinforcement learning policy and a perception module on a shared transformer backbone. This formulation enables adaptability across different tasks by making minimal assumptions about the nature of the underlying POMDP.

- Comprehensive empirical evaluation of two method variants, based on SAC and CrossQ, on classification, volume estimation, and localization tasks, demonstrating that `APPLE` can discover active exploration policies without task-specific heuristics.

## 2 RELATED WORK

Active perception has been studied for both vision and touch Bajcsy (1988); Bajcsy et al. (2018); Bohg et al. (2017); Taylor et al. (2021) and in the context of Applications range from object localization and tracking Yang et al. (2019); Gounis et al. (2024); Tallamraju et al. (2019); Mateus et al. (2022), to scene description Tan et al. (2020), object property identification Boehm et al. (2024), shape estimation Yi et al. (2016); Björkman et al. (2013), UAV navigation Bartolomei et al. (2021), and robotic manipulation De Farias et al. (2021); Xiong et al. (2025a); Dragiev et al. (2013). Many of these works have been formulated using reinforcement learning or non-parametric methods such as Bayesian optimization, with a few imitation-learning-based approaches emerging more recently Yang et al. (2024b); Chuang et al. (2025); Liu et al. (2025); Dai et al. (2022); Xiong et al. (2025b). However, these methods are usually tailored to specific tasks, environments, and objectives, and often assume the agent does not influence the environment through its actions. To our knowledge, there exists no active perception method that has been shown to work on a wide range of tasks, objectives, and environments.

**Active Perception for Tactile Sensing:** Tactile sensing allows robots to infer object geometry, texture, and materials through physical contact, complementing or substituting visual sensing, especially in occluded scenarios. Early tactile sensing systems primarily used simple binary contacts Yousef et al. (2011), whereas recent approaches employ vision-based tactile sensors Yuan et al. (2017); Lambeta et al. (2020); Lloyd & Lepora (2024). These sensors provide high-resolution data useful in complex tasks, including shape reconstruction, texture recognition, and advanced manipulation, such as autonomous page turning Zheng et al. (2022), object reorientation Yin et al. (2023); Qi et al. (2023), scraping Chebotar et al. (2014), and handling deformable objects Bauer et al. (2025). Active tactile perception involves deliberately selecting contact locations and trajectories to optimize information gain during interaction. Previous approaches leverage Gaussian Processes Yi et al. (2016); Björkman et al. (2013) and Bayesian optimization Dragiev et al. (2013); De Farias et al. (2021); Boehm et al. (2024) to efficiently reconstruct shapes, discriminate textures, or identify grasp points. However, these approaches typically rely only on sparse contact points, assume the object is stationary, and often require additional sensing modalities such as vision Björkman et al. (2013). Moreover, most works are tailored to specific tasks. Gaussian process-based methods, such as Yi et al. (2016); Björkman et al. (2013); Dragiev et al. (2013), focus on shape reconstruction, while Boehm et al. (2024) performs texture recognition using vision-based tactile. For comprehensive surveys on tactile manipulation, we refer readers to Li et al. (2020); Yousef et al. (2011).

**Reinforcement Learning in Active Perception:** In active perception, previous works have explored RL-based approaches. In the visual domain, Yang et al. (2019) performs active object search and Tan et al. (2020) generates semantic scene annotations, both use `REINFORCE` to train camera-control policies that improve perception. Related camera-control methods include Cheng et al. (2018), who apply RL to a manipulation task using RCNN-processed visual input, and Dass et al. (2025), who recover a known context variable from visual observations using PPO without memory. Hu et al. (2025) focuses on real-world, vision-based active perception and proposes an RL training recipe that uses privileged sensing and demonstrations to learn deployable viewpoint-selection policies. Active perception for 3D scene understanding has also been studied. Jayaraman & Grauman (2018) learn LSTM-based policies for actively completing panoramic scenes, while Lv et al. (2023) uses a differentiable simulator to select informative viewpoints with attention to sim-to-real transfer. Other works jointly model motor and sensor policies under partial observability, as in Shang & Ryoo (2023), or integrate point-cloud conditioning and distillation for mobile manipulation Uppal et al. (2024).

In the tactile domain, RL-based methods have addressed object shape reconstruction. Particularly, PPO Xu et al. (2022) and DDQN Smith et al. (2021) have been applied to build object shape estimates by selecting informative contact points, with the former assuming a binary tactile sensor in a 2D environment and the latter requiring both vision and touch. By assuming that the object is static, these approaches avoid uncertainty in object pose and maintain explicit shape reconstructions that serve as the policy state. Complementing these, Rajeswar et al. (2022) propose curiosity-driven haptic exploration based on mismatches between visual predictions and tactile observations. Related to our work is the *Recurrent Models of Visual Attention* (`RAM`) Mnih et al. (2014), which uses `REINFORCE` to select sequential image glimpses for MNIST classification via an LSTM policy. Although developed for classification, the idea naturally extends to regression. Building on `RAM`, Fleer et al. (2020) introduce the *Haptic Attention Model* (`HAM`), which learns a control policy

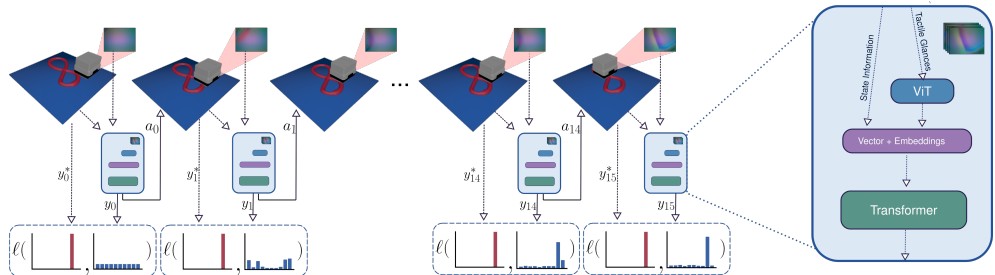

Figure 2: Active perception process in the `APPLE` framework. In this task, the agent's goal is to classify the digit using touch alone. At each step, it receives a tactile reading and state information (e.g., sensor position). A Vision Transformer encodes the tactile input, which is concatenated with state data and processed as a sequence over time by a transformer. At every step, the model outputs a label prediction $y_t$, evaluated against the ground truth $\overset{*}{y}_t$ via a loss function $\ell$, and an action $a_t$ that controls the sensor's next movement.

for a taxel-based tactile sensor to classify four static objects, though training requires millions of interactions. Niemann et al. (2024) extends this with a binary "done" action, again evaluating on tactile classification.

Finally, our classification experiments fit within the broader family of internally rewarded RL (IRRL) methods formalized by Li et al. (2023), which use internal discriminators to produce rewards defined as mutual information between trajectories and labels. While our rewards also depend on the agent's own prediction model, our framework is more general, allowing any differentiable loss rather than targeting mutual information specifically.

## 3 ACTIVE PERCEPTION POLICY LEARNING

Our objective in this work is to develop an active perception method that, unlike prior approaches, is not tied to a particular task or environment. Our guiding principle here is similar to RL. That is, just as RL requires only a reward function, we want to specify a *perception objective* and let the agent learn an appropriate perception policy on its own, without imposing strong task-specific assumptions. On a high level, we frame active perception as a supervised learning problem. The agent's goal is to minimize a loss $\ell(y_t, \overset{*}{y}_t)$ between its current prediction $y_t$ and the ground-truth label $\overset{*}{y}_t$. However, unlike in classical supervised learning, we assume that the agent is not simply presented with a static data point as input, but rather with an interactive environment that it can actively gather data from. E.g., the agent could be presented with an object and has to decide actively how to examine it to extract the information it needs. This perspective defines active perception fundamentally as a sequential decision-making problem embedded within a supervised learning problem. An example of this process is given in Fig. 2. Here, the agent is faced with a classification task, where it must identify a digit from touch alone. Hence, in every step, the agent chooses where to move the sensor while also making a prediction about the class label. The agent's objective is to minimize the loss function $\ell$ throughout this process (illustrated in the bottom row of the Fig. 2). Thus, it must optimize its actions to be as informative as possible.

In the remainder of this section, we formally define our problem and derive the **A**ctive **P**erception **P**olicy **Le**arning (`APPLE`) framework. We propose two variants of `APPLE`, based on `SAC` Haarnoja et al. (2018) and `CrossQ` Bhatt et al. (2019).

### 3.1 PROBLEM STATEMENT

Formally, we define the problem of active perception as a special case of a Partially Observable Markov Decision Process (POMDP). Here, the environment is governed by unknown dynamics $p(\tilde{h}_{t+1}|\tilde{h}_t, \tilde{a}_t)$, where $\tilde{h}_t$ is the hidden environment state and $\tilde{a}_t$ is the action taken at time $t$. The agent then makes observations through the distribution, $p(o_t|\tilde{h}_t)$, where $o_t$ is the observation. In the active perception scenario, the agent's objective is to learn a particular property of the environment, e.g., the class or pose of an object. We assume that the ground truth value $\overset{*}{y}_t$ of this property at time $t$ is part of the hidden state $\tilde{h}_t$ and thus not directly accessible to the agent. Hence, the hidden state

decomposes into $\tilde{h}_t = (h_t, \overset{*}{y}_t)$, where $h_t$ is the remainder of the hidden state without the ground truth property value. Additionally, the agent's action space contains not only control actions $a_t$ but also a current estimate $y_t$ of the desired environment property. In other words, the action space decomposes into $\tilde{a}_t = (a_t, y_t)$, meaning that the agent predicts the desired environment property at every step. As is typical for RL, the action $a_t$ is a control signal, e.g., a desired finger movement, which is communicated to the agent's motor controllers.

The overall reward function $\tilde{r}$ consists of two parts: a differentiable prediction loss $\ell$ and an RL reward $r$. That is, $\tilde{r}(h_t, \overset{*}{y}_t, a_t, y_t) = r(h_t, a_t) - \ell(\overset{*}{y}_t, y_t)$. Here, the prediction loss, $\ell(\overset{*}{y}_t, y_t)$, could, e.g, be a cross-entropy loss in the case of a classification task or the Euclidean distance in the case of a pose estimation task. The RL reward, $r(h_t, a_t)$, does not have to be differentiable or known to the agent. In this work, we only use it to regularize the agent's actions $a_t$. In the following, to simplify the notation, we denote $p\left(\mathbf{h}, \overset{*}{\mathbf{y}}, \mathbf{o}, \mathbf{a}, \mathbf{y}\right) = \pi(\mathbf{y} \,|\, \mathbf{o}) \, p\left(\mathbf{h}, \overset{*}{\mathbf{y}}, \mathbf{o}, \mathbf{a}\right)$, $\pi(\mathbf{y} \,|\, \mathbf{o}) = \prod_{t=0}^{\infty} \pi(y_t \,|\, o_{0:t})$ and $p\left(\mathbf{h}, \overset{*}{\mathbf{y}}, \mathbf{o}, \mathbf{a}\right) = p\left(h_0, \overset{*}{y}_0\right) \prod_{t=0}^{\infty} p(o_t \,|\, h_t) \, \pi(a_t \,|\, o_{0:t}) \, p(h_{t+1} \,|\, h_t, a_t) \, p\left(\overset{*}{y}_t \,\middle|\, h_t\right)$ where $\mathbf{h} \coloneqq h_{0:\infty}$, $\overset{*}{\mathbf{y}} \coloneqq \overset{*}{y}_{0:\infty}$, and so on.

The objective is now to find a policy $\pi(a_t \,|\, o_{0:t})$ for which the expected discounted return is maximized. That is, given the discount factor $\gamma \in [0, 1)$,

$$\max_{\pi} J(\pi) \coloneqq \mathbb{E}_{p\left(\mathbf{h}, \overset{*}{\mathbf{y}}, \mathbf{o}, \mathbf{a}, \mathbf{y}\right)}\left[\sum_{t=0}^{\infty} \gamma^t \tilde{r}(h_t, \overset{*}{y}_t, a_t, y_t)\right]. \tag{1}$$

## 3.2 Optimizing the Active Perception Objective

Let $\ell_{\pi}(\overset{*}{y}_t, o_{0:t}) \coloneqq \mathbb{E}_{\pi(y_t \,|\, o_{0:t})}\left[\ell(\overset{*}{y}_t, y_t)\right]$. Since the agent's predictions $y_t$ do not influence future states, we can rewrite Eq. (1) as

$$J(\pi) = \mathbb{E}_{p\left(\mathbf{h}, \overset{*}{\mathbf{y}}, \mathbf{o}, \mathbf{a}\right)}\left[\sum_{t=0}^{\infty} \gamma^t \left(r(h_t, a_t) - \ell_{\pi}(\overset{*}{y}_t, o_{0:t})\right)\right]. \tag{2}$$

In this work, we assume that the policy $\pi$ is a neural network, parameterized by parameters $\theta \in \mathbb{R}^M$, which allows us to compute a gradient of Eq. (2) and optimize the problem with gradient descent algorithms. Computing the gradient of $J(\pi_{\theta})$ now yields

$$\frac{\partial}{\partial \theta} J(\pi_{\theta}) = \underbrace{\mathbb{E}_{p_{\theta}\left(\mathbf{h}, \overset{*}{\mathbf{y}}, \mathbf{o}, \mathbf{a}\right)}\left[\frac{\partial}{\partial \theta} \ln \pi_{\theta}(\mathbf{a} \,|\, \mathbf{o}) \sum_{t=0}^{\infty} \gamma^t \tilde{r}(h_t, \overset{*}{y}_t, a_t, y_t)\right]}_{\text{policy gradient}} - \underbrace{\mathbb{E}_{p_{\theta}\left(\overset{*}{\mathbf{y}}, \mathbf{o}\right)}\left[\sum_{t=0}^{\infty} \gamma^t \frac{\partial}{\partial \theta} \ell_{\pi_{\theta}}(\overset{*}{y}_t, o_{0:t})\right]}_{\text{prediction loss gradient}}. \tag{3}$$

Intermediate steps can be found in Appendix G. As can be seen in Eq. (3), the gradient of the objective function $J(\pi_{\theta})$ decomposes into a policy gradient and a negative supervised prediction loss gradient.

## 3.3 Deriving APPLE-SAC and APPLE-CrossQ

We use RL-based techniques to estimate the policy gradient in Eq. (3). Here we propose two variations of APPLE based on two actor-critic methods: SAC Haarnoja et al. (2018) and CrossQ Bhatt et al. (2019). SAC is an off-policy RL algorithm that jointly learns a policy and two Q-networks. The Q-networks are optimized to minimize the Bellman residual for the policy, and the policy is optimized to maximize the smaller of the two values predicted by the Q-networks. To stabilize the training of the Q-networks, SAC deploys target networks, which slowly track the actual Q-networks. CrossQ is similar to SAC but drops the target networks and instead uses BatchRenorm Ioffe (2017) layers in the Q-network to stabilize their training. To use either SAC and CrossQ in the active perception setting, we adjust three components:

1. The active perception setting is partially observed. Hence, instead of a state $s_t$, the policy and the Q-networks receive a trajectory of past observations $o_{0:t}$.

2. During the training of the Q-networks, SAC and CrossQ sample transitions and rewards from the replay buffer to compute the Bellman residual. The presence of the prediction loss $\ell_{\pi_{\theta}}$ requires us

to dynamically recompute the total reward when evaluating the Bellman residual, yielding

$$\mathcal{L}_{\text{critic}} = \mathbb{E}_{\mathcal{D}} \left[ \frac{1}{2} \left( Q_\theta(o_{0:t}, a_t) - \left( r_t - \ell_{\pi_\theta}(\overset{*}{y}_t, o_{0:t}) + \gamma \mathbb{E}_{\pi_\theta}[Q_{\bar\theta}(o_{0:t+1}, a_{t+1})] \right) \right)^2 \right].$$

3. During the policy update, the policy gradient is augmented by the prediction loss gradient.

In the following, we denote those `APPLE` variants as `APPLE-SAC` and `APPLE-CrossQ`.

## 3.4 INPUT PROCESSING

We assume that the sequence of past observations $o_{0:t}$ consists of both images and scalar data. In tactile-based active perception, image observations typically correspond to high-dimensional tactile inputs (represented as images), while the scalar data encodes the position of the sensor. As illustrated in Fig. 2, the agent receives this exact combination of tactile images and sensor positions. To efficiently process this data in the policy and Q-networks, we use an architecture similar to the Video-Vision-Transformer (ViViT) Arnab et al. (2021) architecture. First, a Vision Transformer (ViT) Dosovitskiy et al. (2021) is used to generate embeddings for each tactile image. These embeddings are then concatenated with the scalar inputs and processed by a transformer to generate an embedding $m_t$ for each time step. We empirically found that sharing these embeddings across Q-networks $Q_\theta(o_{0:t}, a_t)$, action policy $\pi_\theta(a_t \mid o_{0:t})$ and prediction policy $\pi_\theta(y_t \mid o_{0:t})$ yields better results than training individual representations for each of these components.

CircleSquare      TactileMNIST

TactileMNIST-Volume      Toolbox

Figure 3: Active perception benchmarks on which we evaluate our method. TactileMNIST, TactileMNISTVolume, and Toolbox are tactile perception tasks from the *Tactile MNIST Benchmark Suite* Schneider et al. (2025) where the agent must decide how to gather information with its tactile sensor. CircleSquare and TactileMNIST are classification tasks and TactileMNISTVolume is a regression task, where the agent must determine an object's volume. Toolbox is a pose estimation task, where the agent must determine the 2D pose of the object. All tasks require the agent to gather information actively and are not accurately solvable via random exploration.

## 4 EXPERIMENTS

With `APPLE`, our goal is to answer two questions: *(1)* can we design a general and principled RL-based algorithm that successfully discovers active-perception policies using only a task label and a differentiable loss during training? and *(2)* can such an approach extend across diverse active-perception problems, including both classification to regression without the need to over-design task-specific exploration heuristics? To answer these questions, we focus mainly on the tactile domain, which is particularly suited for active perception due to the local and sparse nature of touch. Thus, we run experiments that span different observation spaces (such as low-dimensional image arrays and higher-dimensional tactile images) and varied downstream prediction goals (including classification and regression). Here, our four evaluated active perception tasks are: *CircleSquare*, *TactileMNIST*, *TactileMNISTVolume*, and *Toolbox* introduced by Schneider et al. (2025) (see Fig. 3). We also compare our approach against the MHSB tactile shape classification task from Fleer et al. (2020). In each task, the agent must actively gather information and jointly learn both a policy and a prediction model. All experiments described in this section are run with 5 random seeds per method, with all models trained from scratch for each seed. In addition to our core configurations, `APPLE-CrossQ` and `APPLE-SAC`, we also evaluate the following baselines:

**(i)** `APPLE-RND`: a random policy baseline sharing the same configuration as `APPLE-SAC`, but not optimizing an action policy. Instead, actions are sampled uniformly at random throughout training. Importantly, while the policy remains random, the perception module is still trained, enabling the model to learn how to interpret tactile observations even without control over the spatial allocation of its haptic glances.

**(ii)** `HAM`: the Haptic Attention Model introduced by Fleer et al. (2020). `HAM` employs a recurrent neural network (LSTM) to integrate tactile observations over time and jointly learns to classify objects while optimizing its exploratory actions through REINFORCE.

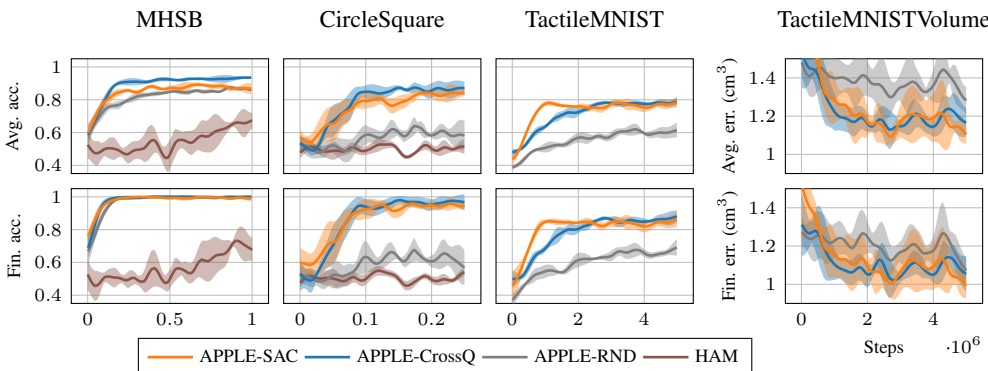

Figure 4: Average and final prediction accuracies for our methods `APPLE-SAC` and `APPLE-CrossQ`, `HAM` Fleer et al. (2020), and `APPLE-RND` across various tasks. *MHSB* refers to the tactile classification task used in Fleer et al. (2020). All methods were trained with 5 seeds. Shaded areas represent one standard deviation. Metrics are computed on evaluation tasks with unseen objects, except for CircleSquare and the MHSB classification task, which have only two or four, respectively.

For evaluation on the classification tasks, we report two complementary metrics. The *average class prediction accuracy* considers the agent's predictions at all steps of an episode and computes the average accuracy across them. The *final class prediction accuracy* only uses the prediction from the final step of each episode, thus showing the model's decision after completing its exploratory sequence. For regression tasks, we report *average error* and *final error*. The hyperparameters of all methods are tuned with `HEBO` Cowen-Rivers et al. (2022). For all methods except `APPLE-CrossQ`, we use one set of hyperparameters, tuned on the TactileMNIST task, for all experiments involving visual tactile inputs and a different set, tuned on the CircleSquare task, for other tasks. For `APPLE-CrossQ`, we use the same hyperparameters, tuned on the TactileMNIST task, for all experiments.

## 4.1 THE MHSB CLASSIFICATION TASK

To enable more direct comparison between our method and `HAM`, we use the benchmark task provided in Fleer et al. (2020). The dataset $\mathcal{D}$ for this benchmark is generated in Gazebo and consists of data from a haptic classification task where a set of four blocks from the Modular Haptic Stimulus Board (MHSB) were arranged in the simulated environment. Each block represents a distinct local shape feature, and data was collected using a simulated Myrmex tactile sensor array that produces 16×16 pressure images upon contact with the block surface. Here, each data point $d \in \mathcal{D}$ consists of the tuple $d = (\boldsymbol{x}, \varphi, \boldsymbol{p})$, where $\boldsymbol{x} \in [-1, 1]$ is the location of the sensor, $\varphi \in \left[-\frac{\pi}{2}, \frac{\pi}{2}\right]$ the angle and $\boldsymbol{p}$ the corresponding normalized pressure (image) array. The agent's goal is to identify the correct object, so at each time step it selects continuous movement actions $a_t = (\boldsymbol{x}, \varphi)$. The closest associated datapoint $d_t \in \mathcal{D}$ is then selected and the agent makes a prediction. Each episode allows the agent to perform 10 touches. For extra details, please refer to Fleer et al. (2020).

In this setting, we evaluate `APPLE-RND`, `APPLE-SAC`, and `APPLE-CrossQ`, with the vision encoder removed to ensure comparability with `HAM`. The first column of Fig. 4 shows the comparison between the different methods on this task. While `HAM` is generally able to solve this task (see Appendix C.5 for a longer run), it requires a large amount of samples to learn an effective policy. After 1M training steps, `HAM` achieves only a final accuracy of 68%, while all of our approaches, including the random baseline, approach 100% after around 250K steps. Additionally, the good performance of the `APPLE-RND` baseline highlights a limitation of this task: due to the discretization of position and angles into 41 bins each, the observation space contains only a total of 1,681 distinct values per class. Agents can quickly learn to memorize all of these values and then solve this task perfectly with just a few random touches. This insight raises questions about `HAM`'s capabilities in learning active perception policies, as `HAM` was only evaluated on this task in Fleer et al. (2020).

## 4.2 THE CIRCLESQUARE TASK

The CircleSquare task has the goal of evaluating active perception in a low-dimensional space. Here, the agent is presented with a 28×28 grayscale image containing either a white circle or square

placed randomly in the field (Fig. 3 top-left). Its goal is to identify the correct object class, but it can only observe a 5×5 glimpse at a time and must explore the image over time. Each episode allows the agent to take up to 16 steps. A color gradient offers directional guidance, but the agent starts without information about the object's location. It selects a continuous movement action $a_t \in [-1, 1]^2$ per step to reposition its glimpse, encouraging learned search strategies over random behavior. Since the agent's glance in this task is small, we again do not use our vision encoder but treat the inputs as a 25-dimensional vector for better comparability to `HAM`, which also processes a flat representation of the input image. In this setting, in addition to the `HAM` baseline, we also compare our `APPLE-SAC` and `APPLE-CrossQ` to the random baseline agent, `APPLE-RND`. As shown in Fig. 4, `APPLE-SAC` and `APPLE-CrossQ` learn to complete the task with similar final prediction accuracies of 97% and 96%, respectively. For a visualization of a policy learned by `APPLE-CrossQ`, refer to Fig. 8. `APPLE-RND`, achieves a 68% accuracy, highlighting the need for active perception in the CircleSquare task. Despite extensive tuning of the learning rate and $\beta$ parameter with `HEBO` and training for 10M steps, we could not find a configuration for which `HAM` reached a prediction accuracy beyond random guessing. For full implementation details, including an ablation with longer training times, where we compare `HAM` with a `PPO` baseline, see Appendix B.1.

### 4.3  TACTILEMNIST CLASSIFICATION

While the CircleSquare environment already presents a non-trivial active perception problem for more generic agents, it remains relatively simple: the input space contains just 25 pixels, there are only two object classes, with a color gradient providing a search direction. In contrast, (visual-) tactile perception add more complexity. First, the input is a high-dimensional image requiring encoding into a latent space. Second, real-world classification tasks often involve many classes with diverse instances. E.g., a robot sorting waste into plastics, glass, and metal must handle objects of various shapes and textures that belong to the same class. Finally, tactile exploration often lacks directional cues, as retrieving an object from a cluttered bag requires systematic search strategies. To investigate this, we evaluate our methods on the *TactileMNIST classification*. Here, an agent uses a GelSight Mini sensor to explore a randomly placed and oriented 3D MNIST digit without prior location knowledge. The goal is to classify the digit within a fixed time budget (see Fig. 3 top-right for a visualization). In this setting, we evaluate `APPLE-SAC` and `APPLE-CrossQ` and compare them with `APPLE-RND`. As this task requires a vision encoder, direct comparison with `HAM` is not possible.

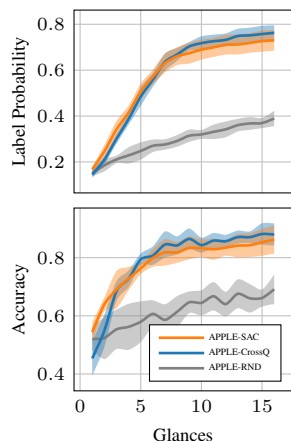

Figure 5: Exploration efficiency of final policies on the TactileMNIST task. Shown are the predicted probability of the correct label (top) and accuracy (bottom) after $N$ glances.

As shown in Fig. 4, both `APPLE-SAC` and `APPLE-CrossQ` reach similar high final prediction accuracies of 87% and 89% on the evaluation task. `APPLE-RND`, however, eventually stagnates at an accuracy of around 74%, highlighting the importance of action selection on this task. The average class prediction accuracy, as shown in Fig. 4 (first column), presents a similar trend, with `APPLE-SAC` achieving 80% and `APPLE-CrossQ` 81%. An additional insight into the agent's performance is given by Fig. 5, which shows how the accuracy and correct label probability of the trained agents over the course of an episode, averaged over multiple episodes. This figure shows that the active agents gather information much quicker and are much more certain about the class label than the random agent. For more details on this task, see Appendix B.2.

### 4.4  TACTILEMNIST VOLUME ESTIMATION

In the *TactileMNISTVolume* task, the agent again uses a GelSight Mini sensor to explore a randomly placed and oriented 3D MNIST digit without prior location knowledge. Unlike the prior classification-based tasks, the objective here is to estimate the digit's volume within a fixed time budget, which makes this task a regression task. A visualization of the TactileMNISTVolume task can be seen in Fig. 3 on the bottom left. To succeed, the agent must simultaneously localize the digit on the workspace and gather sufficient shape information cues to enable accurate volume estimation. For more details about this task, refer to Appendix B.3.

In this setting, we evaluate `APPLE-SAC`, `APPLE-CrossQ`, and `APPLE-RND`, analyzing their ability to perform regression tasks. In the last column of Fig. 4, we show the comparison results for this task. The final average prediction error throughout the task reaches 1.28 cm$^3$ for `APPLE-RND`. `APPLE-CrossQ` and `APPLE-SAC` reach an average prediction error of 1.10 cm$^3$ and 1.16 cm$^3$, respectively. For the final error `APPLE-SAC`, with 0.99 cm$^3$, outperforms both `APPLE-RND` with an error of 1.07 cm$^3$ and `APPLE-CrossQ` with 1.05 cm$^3$. The higher variances and performance fluctuations in Fig. 4 indicate that this task is more challenging than the other tasks. In part, these results might be explained by the fact that neither method was tuned on a regression task. Additionally, estimating the volume of an unknown object requires much more complete information about its shape than classifying it, making both the perception and policy optimization more challenging.

## 4.5 TOOLBOX

In the *Toolbox* task, the agent has to find a wrench on a platform by touch alone. Like TactileMNISTVolume, Toolbox is a regression task, but here, the agent has to predict the 2D position and 1D orientation of the wrench in the workspace. As such, the task consists of two problems: the agent must find the wrench in the workspace and determine its pose. Crucially, as can be seen in Fig. 3 bottom-right, most parts of the wrench are ambiguous in location when touched, and the information of multiple touches has to be combined to make an accurate prediction. For example, when touching the handle, the agent may extract information about the lateral position of the wrench, but does not yet know where it is currently touching the handle longitudinally, and whether the open end is left or right. Hence, to disambiguate the wrench

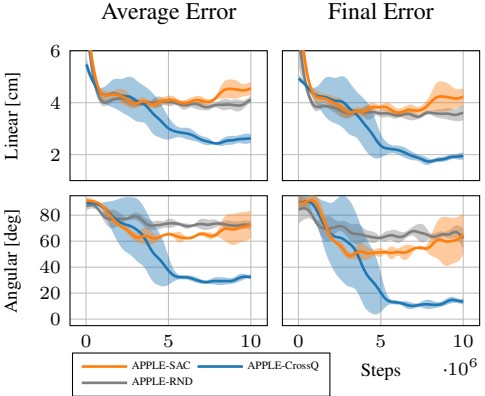

Figure 6: Average and final prediction accuracies for our methods `APPLE-SAC` and `APPLE-CrossQ`, as well as the baseline `APPLE-RND` on the Toolbox task. Each method was trained on 5 seeds for 10M steps.

pose, a strong exploration strategy must include finding one of its ends. Similar to the previous two tasks, the agent moves a GelSight Mini sensor freely in 2D space, constrained by the platform boundaries. Since the platform for this task is larger than that for the other tasks, we allow 64 steps for exploration before the episode is terminated.

The results in Fig. 6 show that `APPLE-CrossQ` reaches a final accuracy of 1.9cm and 13° on average, while `APPLE-SAC` and the random baseline `APPLE-RND` stagnate at much lower accuracies. Throughout the training, `APPLE-CrossQ` learned a sensible exploration strategy, comprised of finding the handle of the wrench and then sliding along it to disambiguate its orientation (see Fig. 12). The low performance of the `APPLE-RND` again highlights the importance of active perception for this task. It is important to note that we again did not tune any of these methods on this task and instead relied on hyperparameters optimized for TactileMNIST. While tuning `APPLE-SAC` on this task can lead to stronger performance, these results indicate that `APPLE-CrossQ` might be more robust w.r.t the choice of hyperparameters. However, more experiments are needed to answer this question definitively. For training and evaluation details, and hyperparameter settings, see Appendix B.4.

## 5 DISCUSSION

The results in Section 4 show that `APPLE` successfully learns exploration strategies across diverse active perception tasks. We evaluated `APPLE` on the CircleSquare classification task and three simulated tactile benchmarks: TactileMNIST (digit classification), TactileMNISTVolume (volume estimation), and Toolbox (pose estimation). In all three tactile benchmarks, the agent must learn an image encoder and refine its exploration policy jointly. The poor performance of the `APPLE-RND` baseline across our tasks confirms the necessity of structured exploration and confirms that our methods learned policies that go beyond random exploration. Notably, `APPLE-CrossQ` retained high performance when switching tasks without hyperparameter tuning, highlighting its robustness.

We compare our method to `HAM` Fleer et al. (2020), which failed to learn an effective strategy even on the Circle-Square toy task, defaulting to predicting the mean class despite extensive tuning using `HEBO`. This points to a fundamental limitation of `HAM` in this setting, namely that `HAM` relies on on-policy RL, which discards samples after a single update, limiting sample efficiency. Further experiments using a `PPO`-based variant of our method (see Fig. 14) underline our hypothesis that on-policy RL is not suited for active perception. In contrast, we employ off-policy methods (`APPLE-SAC` and `APPLE-CrossQ`), enabling sample reuse – a critical factor in active perception, where supervised learning benefits from multiple passes over the same data. Both `APPLE-SAC` and `APPLE-CrossQ` perform comparably on environments they have been tuned on, but `APPLE-CrossQ` has proven more robust to new environments without hyperparameter tuning and offers a clear computational advantage. By avoiding target network updates, it requires roughly half the transformer forward passes during training, leading to a 53% reduction in training time on average, without sacrificing performance. The strong performance of `APPLE`, especially of `APPLE-CrossQ` without retuning, demonstrates its potential as a robust, general framework for active perception.

Particularly, in Fig. 8 and Fig. 12, we show the learned behaviors for our policies in the CircleSquare and Toolbox environments, respectively. In both environments, we can observe that our agents have learned interesting exploration behaviors that ultimately help in solving the task. In CircleSquare, `APPLE` first moves rapidly along the background gradient, which reliably leads toward the object's location. Once it reaches the circle or square, the agent stabilizes its position and keeps the glimpse above the object for the remainder of the episode. More interestingly, in the Toolbox environment, `APPLE-CrossQ` learns to both locate the wrench and disambiguate its orientation. At the beginning of the episode, the agent learns a circular search pattern that efficiently sweeps the workspace. Then, when first encountering the wrench handle, it transitions into a sliding motion along the handle's length. While these patterns are intuitive from a human perspective, for the TactileMNIST and TactileMNISTVolume, the strategies are less clear. In those, the agent typically moves toward the center of the platform early in the episode, as most objects extend into the central region. After initial contact, it often follows local edges or strokes of the digit, but such behaviors vary considerably across digits and seeds. In many cases, the final contacts do not follow a recognizable pattern, suggesting that, in this case, the agent relies more on aggregating many local cues. Overall, these emergent behaviors highlight that `APPLE` can learn different active perception policies, thus adapting its exploration behavior to the demands of each environment with minimal changes to the setup, suggesting a promising step toward more general active perception frameworks.

## 6 CONCLUSION

We introduced `APPLE`, a framework combining reinforcement learning and transformer-based models for active tactile perception. We evaluated it on five benchmarks where `APPLE` consistently outperforms its baselines, underscoring its efficiency in learning exploration policies. Notably, our method requires no hand-crafted heuristics and learns exploration policies in a principled way by minimizing the prediction loss function. The current state-of-the-art method for tactile classification – `HAM` – cannot solve any of these tasks beyond the MHSB task, which it was originally developed for. On the Circle-Square task, `HAM` resorted to always predicting a 50/50 probability for both labels, even after long training, while the other three tasks require an image encoder, which `HAM` lacks. `APPLE`, on the other hand, achieves high performance across all tasks, with the exception of `APPLE-SAC` on the Toolbox task, where it was not tuned on. These results demonstrate `APPLE`'s versatility across diverse tasks, including both classification and regression. Future work will focus on improving the sample efficiency of `APPLE`, extending to more realistic applications such as in-hand pose estimation and texture recognition, and exploring multi-modal integration with vision and touch. Another potential future research avenue will be deploying `APPLE` on a real robotic system. While, in principle, `APPLE` could be applied to real-world tasks as-is, its sample efficiency poses a practical challenge. In other works, sample efficiency issues of RL methods have been addressed through large-scale domain randomization and sim-to-real transfer Bohlinger et al. (2024); Akkaya et al. (2019); Handa et al. (2023). However, their soft gel makes vision-based tactile sensors particularly hard to simulate. Nevertheless, using sim-to-real transfer jointly with accurate soft-body tactile simulation Nguyen et al. (2024) and transferrable representations Yang et al. (2024a) or realistic rendering Chen et al. (2022) are promising avenues towards the application of `APPLE` to real-world tasks.

ACKNOWLEDGMENTS

This work was supported by the German Federal Ministry of Education and Research (BMBF) and the French Research Agency, l'Agence Nationale de Recherche (ANR), through the projects *Aristotle* (Grant no.: ANR-21-FAI1-0009-01), *Chiron* (Grant no.: ANR-20-IADJ-0001-01), and Astérix (Grant no.: ANR-23-EDIA-0002) through the EU's Horizon Europe project *ARISE* (Grant no.: 101135959), the BMBF's projects Robotics Institute Germany (RIG) (Grant no.: 16ME1001), *DEMETER* (Grant no.: 01DR25003), the French national investment priority program's PSPC FAIR WASTE project, and *Genius Robot* (Grant no.: 01IS24083). Furthermore, this work is also supported by the German Research Foundation (DFG, Deutsche Forschungsgemeinschaft) as part of Germany's Excellence Strategy EXC 2050/1 – Project ID 390696704 – Cluster of Excellence *Centre for Tactile Internet with Human-in-the-Loop* (CeTI) of Technische Universität Dresden by the BMBF, and by the German Academic Exchange Service (DAAD) in project 57616814 (SECAI School of Embedded and Composite AI). The computations were conducted on the IAS Compute Cluster.

REPRODUCIBILITY STATEMENT

We took special care to make this work reproducible. The link to the source code can be found on the project page (https://timschneider42.github.io/apple/), and further implementation details are provided in the supplementary material. For transparency, all hyperparameters are listed in Appendix E, and implementation details, including GPU specifications and memory considerations, are provided in Appendix F.

LARGE LANGUAGE MODEL USAGE

A large language model (LLM) was used solely for language editing — polishing phrasing, enhancing readability, and correcting minor typographical errors.

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

## APPENDIX

## A  HAM AS A GENERAL ACTIVE PERCEPTION METHOD

HAM in its original version is an active classification method. Though it is in principle able to use different loss functions than a cross-entropy loss, its reward is defined as a 0-1 reward, yielding 1 for a correct classification and 0 for an incorrect classification. For a regression task, such a reward does not make much sense, as one would have to hand-define thresholds to classify predictions as correct or incorrect. However, we found that on the task it was developed for — the MHSB tactile classification task — using the negative prediction loss directly as a reward yields similar results to the original implementation. With this modification, HAM fits in the APPLE framework, and would, in principle, also be a candidate algorithm for solving the objective in Eq. (1). However, as we show throughout our experiments, HAM's on-policy nature makes it impossible to compete with our off-policy approaches APPLE-SAC and APPLE-CrossQ.

Nevertheless, to allow for a fair comparison, we used this modified version of HAM throughout our experiments. The reasons for this are (a) that otherwise, the APPLE methods and HAM would be evaluated on different rewards, making them less comparable, and (b) that we found in our reproduction study of the work of Fleer et al. (2020), that HAM is, surprisingly, more stable when using the negative prediction loss as a reward.

## B  ENVIRONMENT DETAILS

Here, we detail each of the tasks that were evaluated, with the exception of the MHSB classification task, for which details can be found in Fleer et al. (2020). Thus, we present the details for the CircleSquare 2D classification task, the Tactile MNIST tactile classification and volume estimation tasks, and the Toolbox pose estimation task. These tasks are a part of the TactileMNIST Benchmark Suite, and extra details can be found in Schneider et al. (2025).

### B.1  CIRCLESQUARE TASK

In each CircleSquare episode, the agent receives a 28×28 grayscale image containing either a circle or a square. It can only observe a 5×5 pixel region (glimpse), with the initial location randomized. A color gradient in the background helps guide exploration, but the agent has no access to the object's position. See Fig. 7 and Fig. 8 for an illustration of the CircleSquare task.

Actions $a_t \in [-1, 1]^2$ are mapped to pixel motion as $a_t \cdot 5.6\,\mathrm{px}$ (20% of the image width), allowing smooth movement across the image. We use bilinear interpolation to compute the glimpse values even at non-integer positions. The agent's prediction $y_t \in \mathbb{R}^2$ is interpreted as logits for circle vs. square:

$$p_{\mathrm{circle}}(y_t) = \frac{e^{y_t^{(1)}}}{e^{y_t^{(1)}} + e^{y_t^{(2)}}}, \quad p_{\mathrm{square}}(y_t) = \frac{e^{y_t^{(2)}}}{e^{y_t^{(1)}} + e^{y_t^{(2)}}}.$$

Cross-entropy loss is used for training:

$$\ell(y_t^\star, y_t) = - \sum_{c \in \{\mathrm{circle}, \mathrm{square}\}} \delta(y_t^\star, c) \log(p_c(y_t)).$$

We apply a regularizing reward penalty on the magnitude of each action: $r(h_t, a_t) = 10^{-3} \|a_t\|^2$.

Due to the small input size, we do not use a vision encoder; instead, the input is directly flattened into a 25-dimensional vector. This design choice allows a fairer comparison to HAM Fleer et al. (2020), which also operates on flat image data.

### B.2  TACTILE MNIST CLASSIFICATION

In the Tactile MNIST classification task (see 9), the agent is presented with a 3D model of a high-resolution MNIST digit, placed randomly on a 12×12cm plate. Each digit is up to 10cm in width and height. The agent uses a single simulated GelSight Mini sensor Yuan et al. (2017) to explore the plate

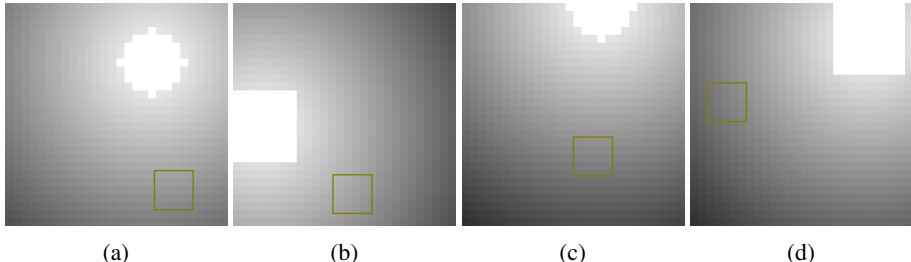

Figure 7: Episode starting conditions of the CircleSquare task. The agent's glimpse and the object (circle (a, c) or square (b, d)) are placed in random locations on the field. Besides the color gradient, the agent receives no information about the object's location on the field.

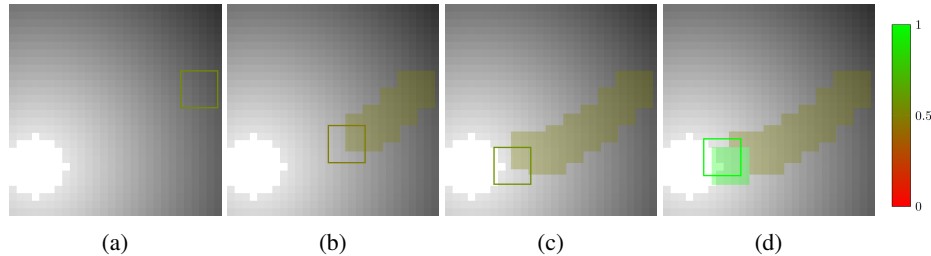

Figure 8: Visualization of a learned `APPLE-CrossQ` policy in the *CircleSquare* task. (a) The agent starts at a random location and uses the color gradient to locate the object. It can only observe a $5 \times 5$ pixel patch. (b) The agent follows the gradient, gradually gathering information. Without full certainty, it predicts a 50/50 probability between classes along the way. Colored boxes show past glances, with color indicating prediction confidence. (c) The agent reaches the object at the corner. (d) Upon confident identification, the agent classifies the object as a square (bright green box) and maintains this prediction in later steps.

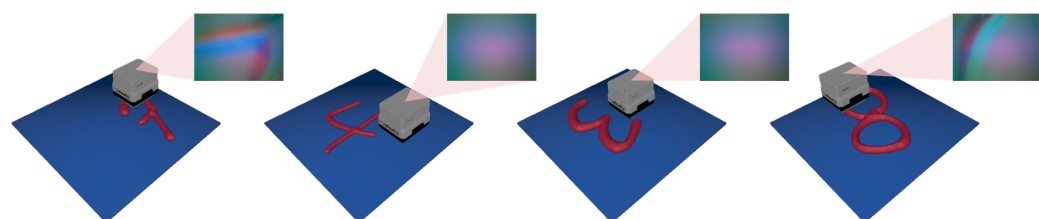

Figure 9: The simulated *Tactile MNIST* classification benchmark Schneider et al. (2025), which we use for evaluating our method. The objective of the Tactile MNIST task is to identify the numeric value of the presented digit by touch only. In every step, the agent decides how to move the finger and predicts the class label. The haptic glance is computed via the Taxim Si & Yuan (2022) tactile simulator.

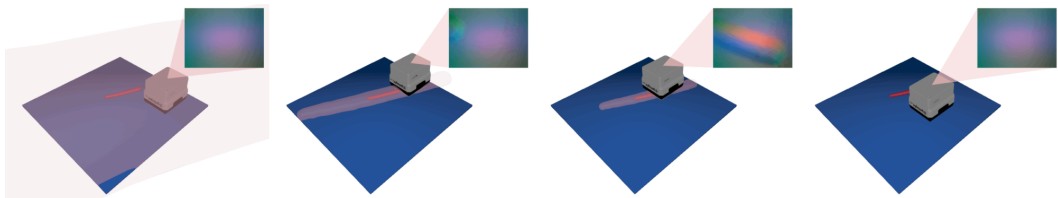

Figure 10: The simulated *Tactile MNIST-Volume* Schneider et al. (2025) task, which we use for evaluating our method. The objective of this task is to estimate a single continuous value representing the volume of a 3D MNIST digit by touch alone. In every step, the agent decides how to move the finger and predicts the volume of the digit. The haptic glance is computed via the Taxim Si & Yuan (2022) tactile simulator, and in red, we show the current volume estimation in a single run.

surface. The sensor outputs a 32×32 pixel image rendered from a depth map by Taxim Si & Yuan (2022). Additionally, the agent receives the 3D position of the sensor as input.

Each episode begins with a randomly selected digit, randomly placed and oriented. The agent has 16 steps to explore and classify the object. At each step, it chooses a movement action $a_t \in [-1, 1]^2$, which corresponds to a translation of up to 1.5cm per axis. The sensor is automatically positioned to maintain a 2mm indentation into the 4mm-thick gel. To simulate the object shifting around when being manipulated by the agent, we apply Gaussian random noise to the position and orientation of the object throughout the episode.

The classification output is a 10-dimensional logit vector. A standard cross-entropy loss is used:

$$\ell(\overset{*}{y}_t, y_t) = -\sum_{c=1}^{10} \delta\left(y_t^*, c\right) \log\left(p_c\left(y_t\right)\right), \quad p_c\left(y_t\right) = \frac{e^{y_t^{(c)}}}{\sum_{i=0}^{10} e^{y_t^{(i)}}}.$$

The reward is used only for regularizing motion: $r(h_t, a_t) = 10^{-3} \|a_t\|^2$.

We train `APPLE-SAC`, `APPLE-CrossQ`, and `APPLE-RND` from scratch (no pre-trained encoders), using 5 random seeds for 5M steps. Hyperparameters are tuned via the `HEBO` Bayesian optimizer. `HAM` Fleer et al. (2020) is not evaluated here as it lacks an image encoder. Evaluation is done using digits not seen during training.

### B.3   TACTILE MNIST-VOLUME

In the Tactile MNIST-Volume task (see Fig. 10), each episode begins with the 3D model of a high-resolution MNIST digit being placed randomly on a 12×12cm plate. The agent explores the scene using a simulated GelSight Mini sensor, identical to that used in the classification variant of the MNIST task. The sensor provides a 32×32 tactile image, and the agent also received its 3D sensor position as input. Actions $a_t \in [-1, 1]^2$ again correspond to a maximum motion of 1.5cm per axis. The sensor maintains a fixed 2mm indentation into the gel on each step. Unlike Tactile MNIST, this is a regression task, and the target in each episode is the volume of the digit, normalized to zero mean and unit variance across all digits.

Each agent – `APPLE-SAC`, `APPLE-CrossQ`, and `APPLE-RND` – is trained from scratch using 5 random seeds for 5M steps. We reuse the hyperparameters optimized on Tactile MNIST without modification to evaluate robustness across tasks. Evaluation is performed on held-out shape configurations, which are not seen during training.

### B.4   TOOLBOX TASK

In the Toolbox task (see 11), the agent has to estimate the pose of a 24cm long wrench that is placed in a uniformly random 2D position and orientation on a 30×30cm plate. Similar to the previous two tasks, the agent explores the scene using a simulated GelSight Mini sensor, providing a 32×32 tactile image, while also receiving the 3D sensor position as input. Actions $a_t \in [-1, 1]^2$ again correspond to a maximum motion of 1.5cm per axis, and the sensor maintains a fixed 2mm indentation into the gel on each touch. To simulate the object shifting around when being manipulated by the agent, we apply Gaussian random noise to the position and orientation of the object throughout the episode.

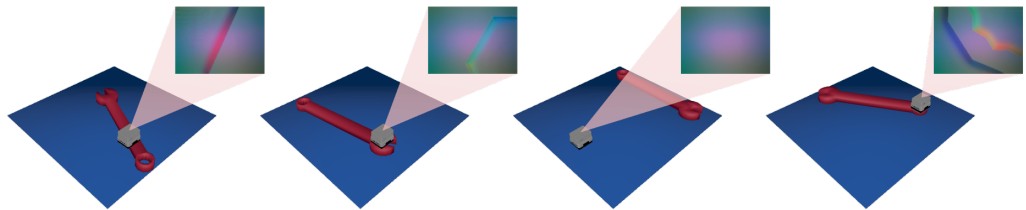

Figure 11: The simulated *Toolbox* task, which we use for evaluating our method. The objective of the Toolbox task is to determine the 2D pose (i.e., 2D position and orientation angle) of the object relative to the platform's center. In every step, the agent decides how to move the finger and predicts the 2D pose. The haptic glance is computed via the Taxim Si & Yuan (2022) tactile simulator.

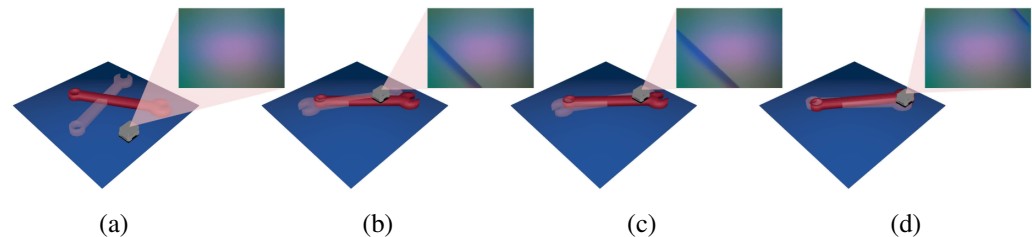

|     (a)     |     (b)     |     (c)     |     (d)     |

Figure 12: Exploration strategy learned by our `APPLE-CrossQ` agent. In the beginning (a), both sensor and wrench start in uniformly random places on the platform. The agent guesses a central position of the wrench (illustrated by the transparent wrench) to minimize error in the absence of any further information. To find the object efficiently, the agent has learned a circular search pattern and therefore quickly locates the object (b). However, the information it currently has is not enough, as the orientation of the wrench is not clear just by touching the handle. Thus, it randomly guesses the wrong orientation, with the open jaw pointing left instead of right. To gather information, it moves along the handle (c) until it finds the open jaw d) and immediately corrects the angle of its pose estimation.

The Toolbox task poses two challenges: finding the object and determining its exact position and orientation. Once the object is found, determining its orientation is still not trivial, as many touch locations only provide ambiguous data. Hence, as shown in Fig. 12, even after the object is found, an exploration strategy for determining its pose must be executed.

Each agent – `APPLE-SAC`, `APPLE-CrossQ`, and `APPLE-RND` – is trained from scratch using 5 random seeds for 10M steps. We reuse the hyperparameters optimized on TactileMNIST without modification to evaluate robustness across tasks.

## B.5    CIFAR-10 CLASSIFICATION TASK

Although omitted in the main paper due to space constraints, we also evaluated our approach on the CIFAR10 task introduced by Schneider et al. (2025). Similar to CircleSquare, the agent moves a small 5x5 pixel glimpse across an image in order to classify it. In contrast to CircleSquare, the agent is presented with 32x32 pixel RGB images from the CIFAR-10 dataset Krizhevsky et al. (2009), which it must classify into 10 classes. See Fig. 13 for an illustration of this task.

Actions $a_t \in [-1, 1]^2$ are mapped to pixel motion as $a_t \cdot 6.4 \, \text{px}$ (20% of the image width), allowing smooth movement across the image. We use bilinear interpolation to compute the glimpse values even at non-integer positions. The agent's prediction $y_t \in \mathbb{R}^{1}0$ is interpreted as logits for the classes:

$$p_c(y_t) = \frac{e^{y_t^{(c)}}}{\sum_{i=1}^{10} e^{y_t^{(i)}}}.$$

Cross-entropy loss is used for training:

$$\ell(y_t^*, y_t) = -\sum_{c=1}^{10} \delta(\overset{*}{y}_t, c) \log\left(p_c(y_t)\right).$$

We apply a regularizing reward penalty on the magnitude of each action: $r(h_t, a_t) = 10^{-3} \|a_t\|^2$.

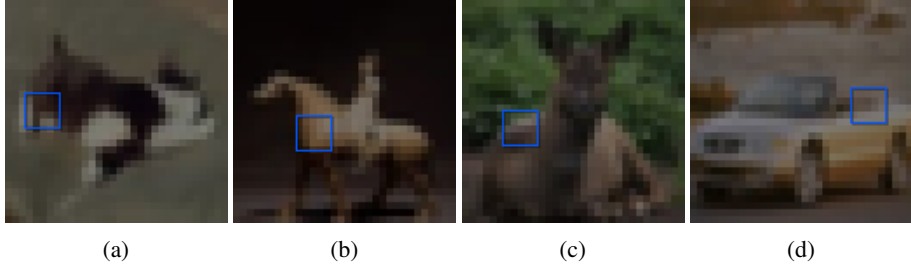

| (a) | (b) | (c) | (d) |

Figure 13: Episode starting conditions of the CIFAR10 task. The agent's glimpse is placed in a random location on the field. It must move the glimpse around in order to learn the class of the displayed image.

Due to the small input size, we do not use a vision encoder; instead, the input is directly flattened into a 25-dimensional vector.

The experimental results for CIFAR10 are presented in Appendix C.6.

## C    FURTHER EXPERIMENTS

In this section, we present additional experimental results that could not be included in the main body due to space constraints.

### C.1    EVALUATING PPO AND HAM ON CIRCLESQUARE

Figure 14 shows an additional experiment on the CircleSquare task Schneider et al. (2025), where we compare HAM to two PPO-based variants, one using HAM's LSTM model (APPLE-PPO-LSTM) and one using our transformer model (APPLE-PPO). Note that, unlike in the experiment shown in Fig. 4 where we stopped training after 250K environment steps, here, we let the training run for 10M steps. Despite the longer training time, HAM fails to achieve a final prediction accuracy that is better than random guessing. The PPO variant using HAM's model (APPLE-PPO-LSTM) achieves a final prediction accuracy of 72% after 10M steps, while the PPO variant with our model (APPLE-PPO) achieves around 79%. In addition to the difference in final performance of APPLE-PPO-LSTM and APPLE-PPO being 7%, APPLE-PPO improves much quicker in the beginning than APPLE-PPO-LSTM. All hyperparameters for this experiment were again tuned using HEBO Cowen-Rivers et al. (2022) and each method was trained from scratch on 5 seeds.

These results, in conjunction with the results shown in Section 4.2, indicate that HAM's issues in solving the CircleSquare task may have two reasons: First, HAM's LSTM model seems to be ill-suited for learning this task, as all other compared methods (APPLE-SAC, APPLE-CrossQ, and APPLE-PPO) each performed worse with HAM's LSTM model than with our transformer model. Second, the fact that PPO and HAM are both on-policy algorithms likely has a negative impact on their sample efficiency, as samples collected from the environment cannot be reused later on. Off-policy algorithms, such as APPLE-SAC and APPLE-CrossQ, on the other hand, store samples over the course of the entire training and revisit them many times, making optimal use of the information gathered during training. We see this effect clearly in Fig. 14, where APPLE-CrossQ outperforms APPLE-PPO by orders of magnitude in sample-efficiency, despite both algorithms using the same model for their policy and class predictions.

### C.2    ABLATION STUDY ON CIRCLESQUARE

To gain an understanding of the effect of the different components of APPLE, we conduct an ablation study on our CircleSquare experiment.

First, we replace the transformer model with an LSTM model. The results shown in Fig. 15 show that, although our model is still learning to solve the task, convergence is much slower compared to the transformer variant of our method, despite extensive tuning of the LSTM baselines with HEBO Cowen-Rivers et al. (2022). These findings are in line with the results from Appendix C.1, in which the transformer-based architecture also outperformed the LSTM architecture when used with a

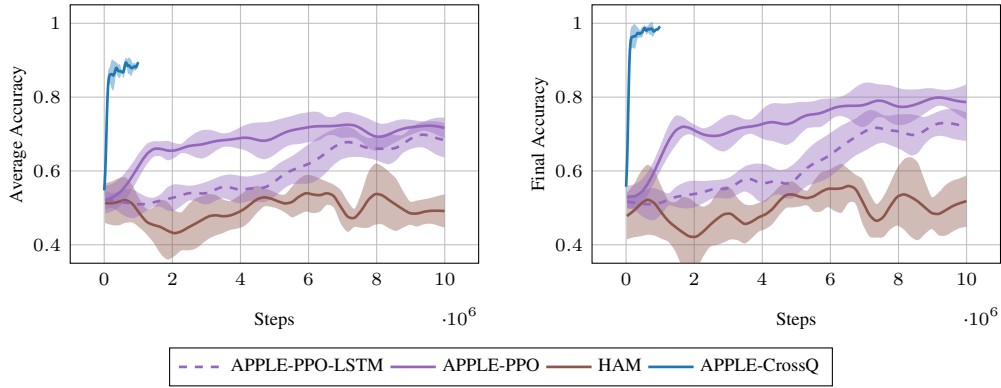

Figure 14: Experiments on the CircleSquare task, comparing `HAM` with two `PPO`-based variants: `APPLE-PPO` and `APPLE-PPO-LSTM`. The difference between `APPLE-PPO` and `APPLE-PPO-LSTM` is that `APPLE-PPO` uses our transformer model, while `APPLE-PPO-LSTM` uses `HAM`'s LSTM model. `APPLE-CrossQ`'s run with 250K steps is shown for reference. The left plot shows the average prediction accuracies, while the right plot shows the final prediction accuracies. Training is terminated after 10M environment steps.

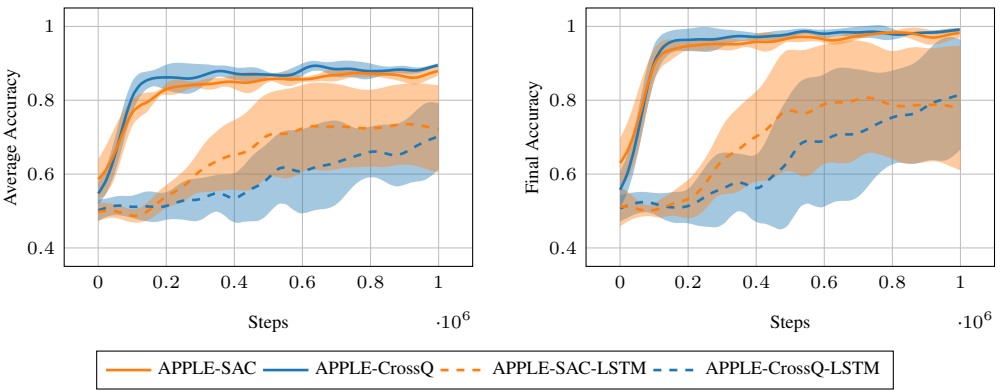

Figure 15: Ablation on the CircleSquare environment, comparing our two `APPLE` variants with modified versions, in which we use an LSTM model in place of `APPLE`'s transformer model. The left plot shows the average prediction accuracies, while the right plot shows the final prediction accuracies. Training is terminated after 1M environment steps.

`PPO` variant of `APPLE`. Hence, we conclude that the choice of model can have a significant influence on `APPLE`'s overall performance.

Second, we investigate the importance of utilizing the target label and the loss function $\ell$ during training. Hence, in this experiment, we just use $\tilde{r}(h_t, \overset{*}{y}_t, a_t, y_t)$ directly as an RL-reward and make $y_t$ become part of the action space. Instead of relying on supervised learning for optimizing $\pi(y_t \,|\, o_{0:t})$, the agent must now rely on its policy gradient and essentially find the optimal $y_t$ via trial and error. We call these variants `APPLE-SAC-PURE-RL` and `APPLE-CrossQ-PURE-RL`, respectively.

As shown in Fig. 16, although the final accuracy of `APPLE-SAC-PURE-RL` briefly peaks to 80%, the absence of target labels and the loss function makes the training of these agents very inefficient and unstable. Since CircleSquare is a fairly simple task, this experiment highlights how challenging it is for RL agents to discover and learn a correlation between their observations and the reward they receive in this setting if no structure is imposed.

Finally, we compare our approach to a variant of itself, which, instead of learning an exploration policy, uses a static grid-search strategy to gather information. This strategy moves the glimpse in a grid pattern across the image, always starting at the closest corner. While this strategy produces better exploration than a random policy, it stagnates at a final accuracy of 73%. The reason for this poor performance is that the time budget of the CircleSquare task is specifically chosen to be too low for exhaustive exploration of the entire environment. Instead, the agent must learn to follow the gradient to efficiently locate and classify the target object.

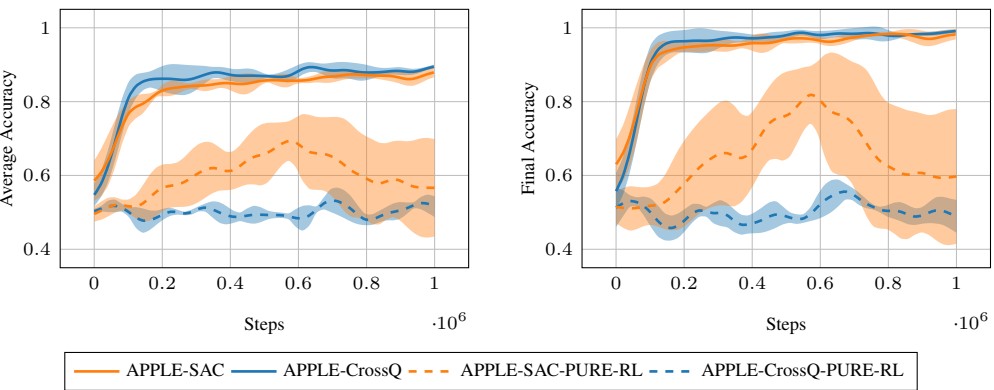

Figure 16: Ablation on the CircleSquare environment, comparing our two `APPLE` variants with modified versions, in which we treat the prediction loss as a regular RL reward, using neither its differentiability nor the target label. Instead, the label prediction becomes part of the regular action space and the agent has to optimize it through its regular policy gradient. The left plot shows the average prediction accuracies, while the right plot shows the final prediction accuracies. Training is terminated after 1M environment steps.

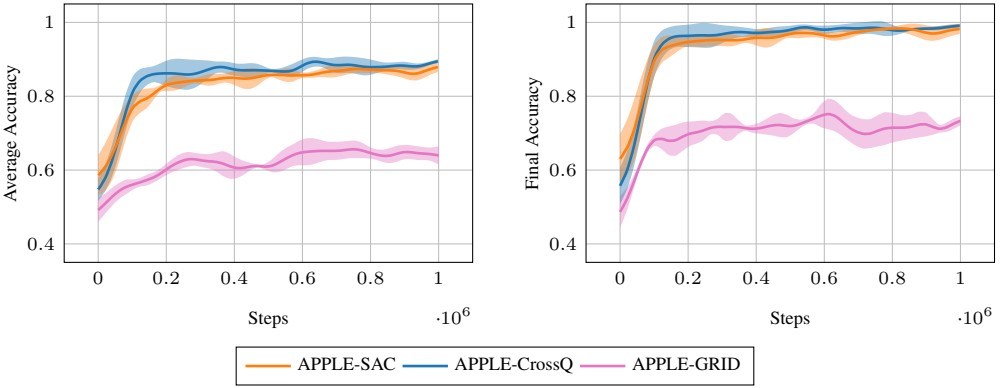

Figure 17: Ablation on the CircleSquare environment, comparing our two `APPLE` variants with a modified version using a heuristic grid-search policy. The grid-search policy searches through the image in a grid pattern, always starting from the closest corner and moving vertically first. Aside from the agent having no control over its actions, everything else is kept the same. The left plot shows the average prediction accuracies, while the right plot shows the final prediction accuracies. Training is terminated after 1M environment steps.

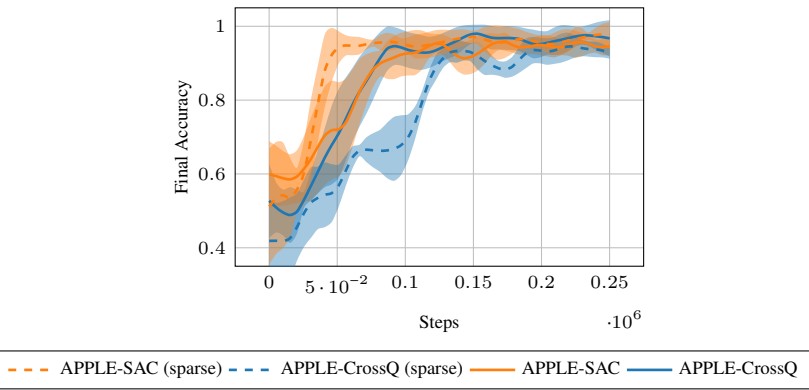

Figure 18: Experiment on a sparse version of CircleSquare, in which only the agent's prediction in the last time step counts. The original learning curve on the non-sparse version of CircleSquare is also displayed for reference. Training is terminated after 1M environment steps.

### C.3    HOW WELL DOES APPLE DEAL WITH SPARSE REWARDS?

In the real world, labels might not always be available for every time step. For example, if a tracking system is used to generate ground-truth data for a tactile pose estimation task, there may be some time steps in the data where the tracking system loses sight of the object. To support this case in our formalism, we can store a boolean variable $w_t \in \{0, 1\}$, indicating whether the object was tracked at time step $t$ ($w_t = 1$) or not ($w_t = 0$) alongside the perception targets $\overset{*}{y}_t$. With a tracking-aware loss function $\ell((\overset{*}{y}, w_t), y_t) := w_t MSE(\overset{*}{y}_t, y_t)$, we can now make sure that only those steps get considered in the prediction loss in which the tracking was successful.

To briefly evaluate the feasibility of using APPLE with sparse rewards, we conducted a small experiment on our CircleSquare environment. In this experiment, we made the reward sparse by using the above loss function and setting $w_t = 0$ for all time steps except the last one, where we set $w_t = 1$. Essentially, the agent receives its perception reward only for the prediction it makes in the final time step. As shown in Fig. 18, we observe that both APPLE variants continue to learn to solve the task, although our CrossQ variant converges slightly more slowly.

### C.4    CAN THE PERCEPTION LOSS HELP IN THE PRESENCE OF ANOTHER DOWNSTREAM TASK?

Our long-term objective with this line of work is to tackle tasks in which active perception is not the primary objective, but rather tasks in which the agent must use active perception to fulfill another objective. One such example is finding and retrieving a specific tool from a toolbox. In this task, the main objective is not to find the tool, but it still has to be found in order to be retrieved. Although we deem these types of tasks to be outside of the scope of this work, we conducted a small experiment to pre-validate the feasibility of using APPLE in this context.

Specifically, we created a variant of the CircleSquare task, in which the agent must stay close to squares and far from circles. We call this variant *CircleSquareHideAndSeek* and implement it by using the following base reward instead of the regular CircleSquare base reward:

$$r(h_t, a_t) = 10^{-3} \|a_t\|^2 + \begin{cases} \|p_t^{\text{agent}} - p_t^{\text{object}}\| & \text{if square} \\ -\|p_t^{\text{agent}} - p_t^{\text{object}}\| & \text{if circle} \end{cases}$$

Hence, to solve this task optimally, the agent must first identify the class of the object and then either remain close or move far away. Here, active perception is only an intermediate step towards solving this task, as knowing the label yields reward only indirectly. In Fig. 19, we compare two variants of our APPLE-CrossQ agent on this task: our regular APPLE-CrossQ agent, which utilizes label and loss-function information, and a variant APPLE-CrossQ-NO-PRED, which just maximizes the base return.

Our results show that the regular agent, which utilizes the label and the loss function, significantly outperforms the agent relying on pure RL. This result suggests that incorporating a supervised

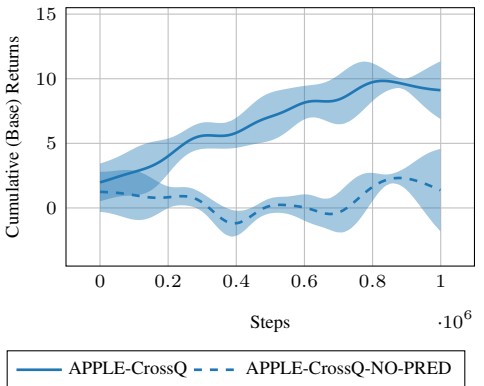

Figure 19: Experiment on the *CircleSquareHideAndSeek* variant of CircleSquare. In this variant, the agent must stay close to squares and avoid circles. We test two variants: `APPLE-CrossQ`, our regular approach, and `APPLE-CrossQ-NO-PRED`, which does not make use of the target label and the loss function. `APPLE-SAC` variants were not included as neither of them learned any meaningful behavior. Training is terminated after 1M environment steps.

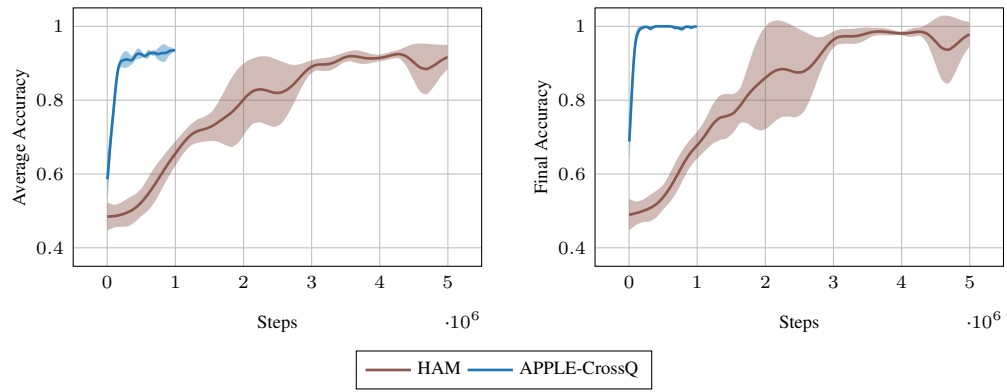

Figure 20: Additional experiments on the MHSB Classification task Fleer et al. (2020), where we let `HAM` run for longer. `APPLE-CrossQ`'s run with 1M step is shown for reference. The left plot shows the average prediction accuracies, while the right plot shows the final prediction accuracies. `HAM` eventually converges to a good policy within 5M steps, reproducing the results of Fleer et al. (2020). Note that the experiment here differs slightly from Fleer et al. (2020), as we use the classification loss as a reward signal for the RL agent instead of a binary reward. We chose to do this modification to allow for a fair comparison to our methods.

learning problem as an inductive bias to guide agents toward discovering relevant information can be beneficial for solving downstream tasks. However, more experiments on more complex environments are needed to investigate whether adding such an active perception bias also brings advantages in broader applications.

## C.5 Longer Evaluation of HAM of the MHSB Task

To validate our implementation of HAM, we conduct a longer experiment on the MHSB task, on which it was originally evaluated. The results of this experiment can be seen in Fig. 20, where we see that HAM eventually converges to good performance. Since these results are similar to those of Fleer et al. (2020), we conclude that our implementation is correct.

## C.6 CIFAR10 Classification Task

In Fig. 21, we present the results of running our two `APPLE` variants, as well as a random baseline, on the CIFAR10 task, described in Appendix B.5. Although the task is similar, CIFAR10 is significantly more challenging than CircleSquare, as the agent now has to deal with much more diverse data and ten labels instead of two.

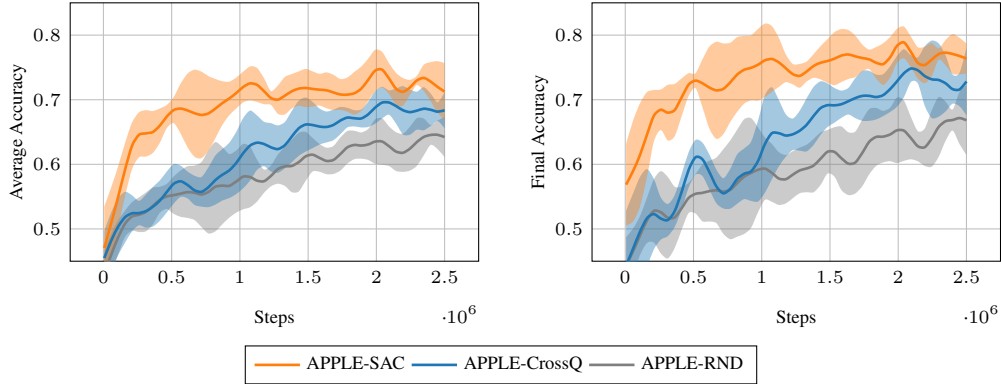

Figure 21: Average and final prediction accuracies for our methods `APPLE-SAC` and `APPLE-CrossQ`, and `APPLE-RND` on the CIFAR10 task. All methods were trained with 5 seeds. Shaded areas represent one standard deviation. Metrics are computed on the test split with unseen images.

As shown in Fig. 21, despite not being tuned for this task, `APPLE-SAC` still performs well, achieving a final accuracy of 76%. `APPLE-CrossQ` converges slightly slower and reaches a slightly lower final accuracy of 73%. Our random baseline reaches a final accuracy of 67% after 2.5M steps.

## D LIMITATIONS

While `APPLE` demonstrates strong performance across various active perception tasks, it also has certain limitations. One of the primary drawbacks is its reliance on large amounts of training data, requiring up to 5M steps for the tactile perception tasks. This high data requirement arises from the combination of a transformer-based architecture and RL-based policy optimization. While this approach enhances the generality of `APPLE`, allowing it to adapt to different tasks without hyperparameter tuning, it comes at the cost of sample efficiency. A promising avenue to solve this issue is leveraging pre-trained transformer models, which could improve sample efficiency by providing useful feature representations. Furthermore, recent advancements in sample-efficient reinforcement learning Nauman et al. (2024); Lee et al. (2025) offer potential alternatives for improving the practicality of `APPLE` in real-world applications. Another limitation and future direction is to explore a more diverse and practical set of tasks. Applications such as object pose estimation, shape reconstruction, or material property inference remain unexplored and could pose additional challenges to our methodology. Moreover, our current experiments use a single tactile sensor, but in principle, the `APPLE` model architecture supports multi-fingered robotic hands and multi-modal perception (e.g., combining vision and touch). However, the practical scalability of `APPLE` to those applications remains an open question, as the increased action and observation space complexity may introduce additional challenges in training efficiency, policy learning stability, and computational demands. Future work will explore these extensions by evaluating `APPLE` on multi-fingered robotic systems and integrating complementary sensing modalities to enhance active perception capabilities.

## E HYPERPARAMETER OPTIMIZATION

For fair comparison between our method and the baselines, we have performed extensive hyperparameter searches with the HEBO Cowen-Rivers et al. (2022) Bayesian optimizer.

**Procedure.** We select the CircleSquare and TactileMNIST classification environments as representative environments on which to tune hyperparameters. Specifically, CircleSquare is used as the representative environment without a vision encoder, and TactileMNIST is used as the representative environment with a vision encoder. We evaluate each candidate configuration by training with a single seed (250K steps on CircleSquare and 2.5M steps on TactileMNIST, except for `HAM` and `PPO`, which are trained for 1M steps on CircleSquare) and measure the episode return averaged across the entire training run. Averaging rewards over time, rather than using final performance alone, ensures

Table 1: Hyperparameters determined by the HEBO Cowen-Rivers et al. (2022) Bayesian optimizer for `APPLE-SAC` and `APPLE-CrossQ`. The *no vision-encoder* configuration was trained on the CircleSquare environment, while the *vision-encoder* configuration was trained on the TactileMNIST environment. Hyperparameters with *Rel.* are relative to the total number of steps throughout the training.

| Hyperparameter | APPLE-SAC | | APPLE-CrossQ | |
|---|---|---|---|---|
| | **no vis.-enc.** | **vis.-enc.** | **no vis.-enc.** | **vis.-enc.** |
| Optimizer type | ADAMW | ADAMW | ADAMW | ADAMW |
| Learning-rate (actor) | $5 \cdot 10^{-5}$ | $5 \cdot 10^{-4}$ | $1 \cdot 10^{-5}$ | $3 \cdot 10^{-4}$ |
| Learning-rate (critic) | $5 \cdot 10^{-4}$ | $5 \cdot 10^{-5}$ | $1 \cdot 10^{-4}$ | $6 \cdot 10^{-5}$ |
| LR-schedule (both) | none | cosine-decay | none | none |
| Rel. LR cosine warm-up (both) | N/A | 0.15 | N/A | N/A |
| Initial UTD | 0.75 | 0.25 | 5.0 | 0.25 |
| Final UTD | 4.0 | 1.5 | 0.5 | 3.5 |
| Rel. UTD warm-up | 0.9 | 0.4 | 0.3 | 0.45 |

that sample efficiency is taken into account: two hyperparameter sets achieving the same final return may differ greatly in how quickly they reach that level of performance.

**Search space.**   Because Bayesian optimization scales poorly with dimensionality, we restrict the search to hyperparameters we found most impactful for performance. All methods are tuned for learning rate, learning-rate schedule (none, cosine decay, linear), schedule parameters (e.g., warm-up steps), and optimizer choice (ADAM, ADAMW, SGD). For off-policy `APPLE` methods, we additionally tune the update-to-data (UTD) schedule: initial and final UTD ratios and the number of warm-up steps.

**Findings.**   Despite an extensive search, we were unable to identify hyperparameters yielding competitive performance for `HAM` on CircleSquare. In contrast, the search provided valuable insights for `APPLE-SAC` and `APPLE-CrossQ`. Although both achieved comparable performance on their tuning tasks, `APPLE-CrossQ` demonstrated substantially greater robustness when transferred to unseen environments (Section 4). Interestingly, applying `APPLE-CrossQ`'s vision-encoder hyperparameters to CircleSquare produced no measurable degradation in performance. To simplify evaluation, we therefore adopt the vision-encoder configuration for all environments in subsequent experiments.

The final hyperparameter settings selected by HEBO are summarized in Table 1.

## F   IMPLEMENTATION DETAILS

The implementations of `APPLE`, `APPLE-PPO`, and `HAM` are built on JAX Bradbury et al. (2018) with the Flax framework Heek et al. (2024), and use Hugging Face transformers Wolf et al. (2020). For performance, the training loop is fully JIT-compiled, and environment interactions are handled via host callbacks—maximizing throughput at the expense of some implementation flexibility.

**Replay buffer design.**   A common bottleneck in deep learning arises from the transfer of data between VRAM (GPU) and RAM (CPU). To minimize this overhead, we keep the replay buffer in VRAM, so that host-device communication is limited to stepping the environment and logging. The drawback is reduced capacity for environments with vision inputs, since VRAM is smaller than RAM. On Nvidia RTX A5000 GPUs (24GB VRAM), storing downscaled $32 \times 32$px visual inputs allows for roughly 3M transitions before memory is exhausted. Consequently, for vision-encoder configurations (i.e., the Tactile MNIST classification and volume estimation tasks as well as the Toolbox localization task), we set the replay-buffer size to 3M, whereas for non-vision-encoder configurations, we use a replay-buffer size equal to the total number of environment steps.

**Hardware setup.**   All experiments were run on Nvidia RTX A5000 or RTX 3090 GPUs with the hyperparameters from Table 1. For vision-encoder configurations, we dedicate one GPU per run. A single run of 5M training steps takes about 40–50 hours, depending on the algorithm.

**Parallelization.** Non-vision-encoder configurations demand less VRAM, enabling multiple runs to share a GPU. On a single RTX A5000/3090, we can accommodate up to 28 parallel runs, depending on the environment and algorithm. As wall-clock runtime then depends heavily on the number of concurrent runs, reporting averages is not meaningful. Nevertheless, we observe that `HAM` and `PPO` typically run about four times faster than `APPLE`.

## G    DETAILED DERIVATION OF EQ. (3)

Due to space constraints, we omitted intermediate steps in the derivation of Eq. (3) in the main paper and instead note them here:

$$
\begin{aligned}
\frac{\partial}{\partial\theta}J(\pi_\theta) &= \frac{\partial}{\partial\theta}\mathbb{E}_{p_\theta\left(\mathbf{h},\overset{*}{\mathbf{y}},\mathbf{o},\mathbf{a}\right)}\left[\sum_{t=0}^{\infty}\gamma^t\left(r(h_t,a_t)-\ell_{\pi_\theta}(\overset{*}{y}_t,o_{0:t})\right)\right] \\
&= \frac{\partial}{\partial\theta}\int p_\theta\left(\mathbf{h},\overset{*}{\mathbf{y}},\mathbf{o},\mathbf{a}\right)\sum_{t=0}^{\infty}\gamma^t\left(r(h_t,a_t)-\ell_{\pi_\theta}(\overset{*}{y}_t,o_{0:t})\right)d(\mathbf{h},\overset{*}{\mathbf{y}},\mathbf{o},\mathbf{a}) \\
&= \int \frac{\partial}{\partial\theta}p_\theta\left(\mathbf{h},\overset{*}{\mathbf{y}},\mathbf{o},\mathbf{a}\right)\sum_{t=0}^{\infty}\gamma^t\left(r(h_t,a_t)-\ell_{\pi_\theta}(\overset{*}{y}_t,o_{0:t})\right) \\
&\quad + p_\theta\left(\mathbf{h},\overset{*}{\mathbf{y}},\mathbf{o},\mathbf{a}\right)\frac{\partial}{\partial\theta}\sum_{t=0}^{\infty}\gamma^t\left(r(h_t,a_t)-\ell_{\pi_\theta}(\overset{*}{y}_t,o_{0:t})\right)d(\mathbf{h},\overset{*}{\mathbf{y}},\mathbf{o},\mathbf{a}) \\
&= \underbrace{\mathbb{E}_{p_\theta\left(\mathbf{h},\overset{*}{\mathbf{y}},\mathbf{o},\mathbf{a}\right)}\left[\frac{\partial}{\partial\theta}\ln\pi_\theta(\mathbf{a}\,|\,\mathbf{o})\sum_{t=0}^{\infty}\gamma^t\tilde{r}(h_t,\overset{*}{y}_t,a_t,y_t)\right]}_{\text{policy gradient}} - \underbrace{\mathbb{E}_{p_\theta\left(\overset{*}{\mathbf{y}},\mathbf{o}\right)}\left[\sum_{t=0}^{\infty}\gamma^t\frac{\partial}{\partial\theta}\ell_{\pi_\theta}(\overset{*}{y}_t,o_{0:t})\right]}_{\text{prediction loss gradient}}.
\end{aligned}
$$

## H    CONNECTION TO CURIOSITY-BASED INTRINSIC REWARD METHODS

Since formulation in Eq. (2) somewhat resembles the augmented rewards commonly used in curiosity-based intrinsic reward RL methods, such as RND Burda et al. (2019) or ICM Pathak et al. (2017), one might wonder what the connection between `APPLE` and those methods is. Fundamentally, `APPLE` solves a different problem than curiosity-based intrinsic reward methods. Intrinsic reward methods typically focus on enhancing policy learning in an MDP setting by providing intrinsic exploration bonuses for under-explored areas of the state space. Hence, they are concerned with letting the agent collect rich experiences to facilitate efficient policy learning. Crucially, the MDP the agent faces during training is, because of the augmentation with the intrinsic reward bonus, different from the one it faces during evaluation.

`APPLE`, on the other hand, is concerned with learning hidden properties of the environment within an episode. Although our formulation may resemble the augmented rewards of intrinsic motivation methods, it operates on a fundamentally different level of exploration. In our case, the extraction of information is the objective posed by the task and not a surrogate used to learn better policies.

On a high level, one could say that intrinsic reward methods utilize exploration to learn the optimal policy parameters, while `APPLE` uses exploration to learn about the hidden state of the POMDP it operates in. Hence, the exploration procedure of intrinsic reward methods occurs across episodes, while the exploration procedure of `APPLE` takes place within each episode. These two concepts are fully orthogonal, and one could indeed combine ICM or RND with APPLE in an attempt to speed up its learning progress.

