# OpenReview forum: "APPLE: Toward General Active Perception via Reinforcement Learning"
_ICLR.cc/2026/Conference — ICLR 2026 Poster_

### Official Review · Reviewer_yWQL · 2025-10-22

**Soundness:** 2
**Presentation:** 3
**Contribution:** 2
**Rating:** 2
**Confidence:** 4

**Summary:**

The APPLE paper introduces a unified framework for active perception that jointly trains a sensing policy and a prediction model within the learning process. Instead of treating perception and control as separate modules, the method combines them through a single objective that encourages both accurate predictions and efficient sensing. The approach is implemented using standard off-policy reinforcement learning methods and evaluated on simulated visual and tactile tasks. The results show consistent improvements over random and simple baseline policies.

**Strengths:**

**Strengths**

**1. Clear Motivation and formulation:** Presents a simple, mathematically sound loss-augmented return that ties prediction and control under a single objective. Theoretical formulation is also explained clearly and is easy to follow.

**2. Good Writing:** The paper is clear, concise, and well-organized. Its explanations of tasks and evaluation approach are logically structured, facilitating straightforward comprehension of the methodology and results.

**3. Implementation clarity:** Detailed reporting of prediction rewards, loss and hyperparameters are provided, and it seems easy to reproduce.

**Weaknesses:**

**Weaknesses**

**1. Overstated Claims:** The paper repeatedly suggests that APPLE is sensor-agnostic and can integrate data from “diverse sensor inputs without task-specific modifications” (Line 88). However, every reported experiment uses only a single simulated sensor type: the tactile tasks (TactileMNIST, Volume, Toolbox) rely on the GelSight Mini, while the visual task (CircleSquare) uses a synthetic 5 × 5 pixel grid rather than an actual camera. For justification, the authors must provide more experiments on diverse settings.

**2. Novelty:** The paper’s conceptual foundation is not particularly novel, as it builds upon long-established principles in active perception and intrinsic exploration. A broad range of prior work spanning attention-based models, curiosity-driven reinforcement learning, and active sensing  has already linked exploratory behavior to prediction-based or feedback-driven learning signals. These approaches share a common theme: agents act to maximize informational or predictive value, whether expressed through task accuracy, novelty, or uncertainty reduction. APPLE remains situated within this same conceptual framework, differing mainly in its operationalization, substituting intrinsic prediction error with a supervised task loss embedded in the reward. While the formulation is elegant and well-executed, it represents a refinement of existing paradigms rather than a substantive theoretical advancement.

**3. Experimental Evaluations:** The experimental evaluation remains simulation-bound and lacks any real-world validation, making the conclusions about generality largely speculative. All tactile experiments rely on simulated tactile images rather than data from a physical sensor, so issues such as contact noise, latency, or sensor drift are never addressed. The supposedly “general” Toolbox benchmark is limited to a single rigid object (a wrench) and one fixed workspace, offering no evidence of cross-object or cross-material generalization, which is an essential aspect for active perception in robotics. Moreover, there are no multi-object or multi-tool variations to demonstrate robustness to geometry or frictional diversity, and no tests in cluttered environments. This narrow scope undercuts the paper’s broader claims of task and sensor generality.

**Questions:**

1. The paper emphasizes multi-sensor and task-agnostic capabilities, yet all experiments are single-sensor and per-task. Could you clarify what evidence supports those claims?

2. Can the framework handle sensor noise or real-world uncertainty (e.g., drift, latency, or stochastic contacts)?

3. How does APPLE differ algorithmically from curiosity-based intrinsic reward methods such as ICM or RND, beyond replacing intrinsic prediction error with supervised task loss?

---

> ### Author Response · Authors · 2025-11-22
>
> Thank you for the comments and for taking the time to review our work.
> We appreciate that you found the paper’s motivation and mathematical formulation elegant and well-executed and that it is well-presented.
> Below, we address all the concerns that have been raised.
>
> ## On the generality of APPLE
> We would like to clarify that **we do not claim that APPLE solves all problems that fall into our definition of active perception in Section 3.1, let alone arbitrary active perception problems in general**.
> We clarified the scope of this work in the introduction to ensure that the framing of this work is not misleading.
>
> It is true that APPLE was designed with generality in mind, making as few assumptions about the problem as possible, similar to how RL algorithms try to be as problem-agnostic and general as possible.
> This principle contrasts with many prior active perception methods, which are often fundamentally tied to specific sensor types or environments (we added more details to our related work section, where we discuss these approaches).
> The fundamental research question we are asking is: _Can we design an active perception algorithm that learns to solve problems without the injection of significant prior problem knowledge into the engineering process?_
> We do neither intend to claim that APPLE learns generalist exploration policies, nor that our experiments suffice to show APPLE working across vast domains of active perception problems.
> Yet, our experiments show that within the simulated tactile setting chosen for evaluation in this work, APPLE learns effective exploration strategies for different tasks (shape classification, volume estimation, and pose estimation) and sensors (pixel glimpse, GelSight Mini, the Myrmex taxel array, and proprioception).
> To further diversify the set of benchmark problems, we have now added an experiment for image classification on the CIFAR-10 dataset, where an agent has a limited window of the image.
>
> Our long-term vision for this line of work is to develop a method that enables active perception across a wide range of domains, including vision, tactile, and possibly audio.
> To achieve this goal, we believe that methods are needed that make as few assumptions as possible about the problem, and we argue that our unified formulation is an important step in this direction.
> However, at this point, we also acknowledge that further research and experimentation are necessary to fully achieve this long-term goal.
> We discuss the current limitations of APPLE in Appendix D.
>
> Regarding the statement in Line 88, we state that APPLE's transformer-based architecture allows for the accommodation of diverse sensor inputs.
> We want to emphasize that **we do not claim our method to be fully sensor-agnostic**.
> Currently, our implementation supports image data through the ViT encoder and vector data, which we pass through a dense MLP to create embeddings.
> We argue that most data stemming from robotic sensors falls into one of these two categories, which is the basis for our claim that our architecture can accommodate diverse sensor inputs.
> Of course, there are exceptions, such as point cloud or (scene-)graph data, which are not easily represented as vectors or images, and for which specialized encoders are needed to obtain embeddings.
> While we consider the processing of such data to be beyond the scope of this work, we would like to point out that numerous studies have focused on computing embeddings for point clouds [1-4] and graphs [5-6].
> Integrating those into APPLE is definitely a promising avenue for expanding its applicability.
>
> Should there be statements in the paper that are unclear with regard to the scope of this work or to APPLE's generality, we are happy to revise and clarify them.

---

> ### Author Response · Authors · 2025-11-22
>
> ## Novelty
>
> Regarding the comment that our approach is merely a refinement of existing frameworks, we would like to clarify that while APPLE is indeed situated within active perception, this area still has considerable potential for growth.
> By active perception, we mean that we provide a framework where perception and control are tightly coupled, often leading to exploratory action.
> However, we believe our approach constitutes more than a small refinement of existing paradigms.
> First, to the best of our knowledge, our framework provides the first unified and explicitly task-driven formulation of active perception within a POMDP framework, where the objective is grounded directly in supervised prediction loss, rather than heuristic intrinsic uncertainty reduction.
> Our goal here is not simply to reduce uncertainty, but to learn exploration policies (or active perception policies) that solve a given task while gathering information, with the task being defined by the prediction loss.
> This design encourages the emergence of meaningful exploratory behaviors, such as the strategy shown in Figure 12, where the agent follows the edges of the wrench to disambiguate its pose, without directly optimizing for uncertainty minimization of overall uncertainty.
>
> Second, we reiterate that prior work in active perception typically introduces task-specific heuristics or handcrafted exploration strategies.
> In contrast, our formulation intentionally minimizes assumptions about the task structure, allowing it to be applied to different active perception problems without modification.
> While this is not applicable to every conceivable task, we believe it to be a first step towards that goal.
>
> Third, regarding methodology, we note that APPLE represents, to our knowledge, the first application of off-policy reinforcement learning combined with a transformer-based architecture in a partially observed active perception setting.
> Although transformers have been successful in other domains, we demonstrate their effectiveness in this context and show that they offer both architectural and practical advantages over prior approaches.
> Finally, APPLE supports both vector-based and vision-based inputs, and its ability to handle non-static scenes, in which objects may move, further demonstrates a robustness that we have not seen in previous approaches.
>
> We have clarified these contributions in the introduction of the revised manuscript, but to summarize, APPLE contributes with:
>
> - A principled unified formulation of active perception as a POMDP optimized for supervised prediction loss rather than heuristic objectives.
> - The first application of off-policy RL with a transformer-based model in this domain.
> - A general, assumption-light framework without handcrafted heuristics, capable of handling multiple tasks and sensors.
>
> Furthermore, we distinguish our approach from curiosity-based intrinsic reward RL methods in the final paragraph of this response.
>
> Finally, if there are any relevant works we may have missed that are close enough to raise concerns about novelty, we would be glad to discuss them and explain how our work differs.
>
>
> ## Clarification regarding experimental evaluations
> As noted in our limitations section, APPLE currently requires substantial amounts of interaction data, making on-robot training impractical at this stage, and sim-to-real transfer is outside the scope of this work.
> Our goal here is to introduce and validate a formulation for active perception, rather than to demonstrate a full real-world deployment pipeline.
> Regarding the concern that the Toolbox benchmark alone does not demonstrate cross-object or cross-material generalization, we agree.
> We want to emphasize that **we do not claim the Toolbox task to be “general” on its own**.
> The Toolbox is one among several environments used to evaluate APPLE.
> Our evaluation in the paper includes:
> - The TactileMNIST, with more than 13k digits across 10 classes, was tested under both classification and volume estimation settings.
> - The MHSB (shape classification) task, using a different tactile sensor, across four distinct 3D shapes.
> - The mentioned Toolbox task for pose estimation.
> - The Circle–Square task, a smaller toy example for vector data (the input glimpse is flattened before processing), where we have to classify two shapes.
>
> Additionally, we now also provide a 2D image classification task on the CIFAR-10 dataset.
> Together, these environments reflect multiple objects, multiple tasks, multiple modalities (tactile and visual), and multiple prediction objectives (classification, regression, and pose estimation), showing that our method can be applied in different contexts.

---

> ### Author Response · Authors · 2025-11-22
>
> ## Handling sensor noise
> The focus of this paper generally lies on the algorithmic aspect of finding a method that is capable of learning exploration policies without making strong assumptions about the nature of the task.
> While we acknowledge the importance of dealing with sensor drift, latency, and stochasticity for real-world deployment, we deem an in-depth investigation of these aspects to be out of scope of this work and plan to address them in future work.
>
> Nevertheless, even in the current version, the agent is already facing some of the challenges you mention in its tasks.
> First, the objects in all our experiments move around randomly by a couple of millimeters in every step, which cannot be predicted by the agent.
> Hence, contact with the object is already stochastic to some degree.
> Second, we have conducted a classification experiment on CIFAR-10, which contains natural images, thereby presenting the agent with the challenge of dealing with sensor noise.
> Generally, because APPLE relies fundamentally on deep-learning RL techniques, we expect similar challenges to arise as in other RL methods, where sensor drift and stochasticity can be reasonably dealt with, as long as they stay within the training distribution.
>
>
> ## Differences to curiosity-based intrinsic reward methods
>
> APPLE solves a fundamentally different problem than curiosity-based intrinsic reward methods such as ICM or RND, and is, as such, not really comparable.
> Intrinsic reward methods, including ICM and RND, typically focus on enhancing policy learning in an MDP setting by providing intrinsic exploration bonuses for under-explored areas of the state space.
> Hence, they are concerned with letting the agent collect rich experiences to facilitate efficient policy learning.
> Crucially, the MDP the agent faces during training is, because of the augmentation with the intrinsic reward bonus, different from the one it faces during evaluation.
>
> APPLE, on the other hand, is concerned with learning hidden properties of the environment within an episode.
> Although our formulation may resemble the augmented rewards of intrinsic motivation methods, it operates on a fundamentally different level of exploration.
> In our case, the extraction of information is the objective posed by the task and not a surrogate used to learn better policies.
>
> On a high level, one could say that intrinsic reward methods utilize exploration to learn the optimal policy parameters, while APPLE uses exploration to learn about the hidden state of the POMDP it operates in.
> Hence, the exploration procedure of intrinsic reward methods occurs across episodes, while the exploration procedure of APPLE takes place within each episode.
> These two concepts are fully orthogonal, and one could indeed combine ICM or RND with APPLE in an attempt to speed up its learning progress.
>
> We have clarified the differences between APPLE and intrinsic reward methods in Appendix H.
>
> ## References
> [1] Guo, Meng-Hao, et al. "Pct: Point cloud transformer." Computational visual media 7.2 (2021): 187-199. \
> [2] Zhao, Hengshuang, et al. "Point transformer." Proceedings of the IEEE/CVF international conference on computer vision. 2021. \
> [3] Park, Jinyoung, et al. "Self-positioning point-based transformer for point cloud understanding." Proceedings of the IEEE/CVF conference on computer vision and pattern recognition. 2023. \
> [4] Zeng, Jiahao, Decheng Wang, and Peng Chen. "A survey on transformers for point cloud processing: An updated overview." IEEE Access 10 (2022): 86510-86527. \
> [5] Cai, Hongyun, Vincent W. Zheng, and Kevin Chen-Chuan Chang. "A comprehensive survey of graph embedding: Problems, techniques, and applications." IEEE transactions on knowledge and data engineering 30.9 (2018): 1616-1637. \
> [6] Xu, Mengjia. "Understanding graph embedding methods and their applications." SIAM Review 63.4 (2021): 825-853.

---

### Official Review · Reviewer_LSx3 · 2025-10-25

**Soundness:** 2
**Presentation:** 2
**Contribution:** 2
**Rating:** 4
**Confidence:** 3

**Summary:**

The authors propose to solve active perception tasks, where the agent  interacts with the environment to collect features for prediction. The work is focused on tactile manipulation tasks where the robot must feel objects to determine their class. They use a transformer based architecture and optimize a reward function that minimizes prediction error. They show results on simulated benchmarks where the agent explores via touch sensor or limited field of view to determine an object of interest. They outperform a random policy and an LSTM policy.

**Strengths:**

The proposed objective and architecture is sensible for solving active perception tasks in the tactile sensing domain, although I have some issues with the framing of it as a general framework.

Active perception for tactile sensing is a niche area with lots of potential for growth, so this paper is very timely.

The paper is well written and provides good background on tactile sensing and active perception.

**Weaknesses:**

## Claim of generality of method is misleading
The authors claim that this active perception method is not tied to any particular task, and is "general". This is misleading and needs to be revised.

The method proposes an additional reward term, which is a differentiable prediction term. First, requiring the term to be differentiable reduces the amounts of tasks this method is applicable to. Next, adding an additional prediction term introduces tuning complexity and the danger of reward misspecification. This objective would not work in tasks where it is hard to specify the state (no access to pose or  state).

Finally, perhaps most importantly, the authors only evaluate in tasks where the task success is defined as correctly perceiving the state. But in most realistic tasks, perceiving the state is only an intermediate step towards solving the task. And the degree to which we need to perceive the state to solve the task is often quite low.

There is an approach that bypasses all of these limitations.  Maximizing the return with RL is the most general approach. If the task demands active perception in the optimal policy, then RL will discover it to maximize reward. Of course, this can be intractable in many cases, so we need additional mechanisms, like the reward term proposed, to improve learning. But this reduces generality.

I would like the authors to reframe their claims here, perhaps saying this is a good way of solving a particular class of active perception tasks where the goal is to discover a state, without overstating the generality.

## Lack of baselines
I think there could be more baselines than just a random policy and a LSTM policy. For example, what about a hardcoded raster-scanning policy? Or some policy that uses a heuristic for exploration / information gain, etc. These simple approaches would be considered first for a roboticist.


## Framing of background work
I have an issue with the background section. It has a few sentences that I disagree with.
>In the context of active perception, a few works have explored RL as an option.
> However, these methods are usually tailored to specific tasks, environments, and objectives, and often assume the agent does not influence the environment through its actions. To our knowledge, there exists no active perception method that has been shown to work on a wide range of tasks, objectives, and environments."

These sentences give the feeling that active perception is under-explored, and current methods are unsatisfactory in generality and performance. But this is not true.

A quick search on the web for most recent papers on active perception, e.g.  "CoRL 2025 active perception" or "Neurips 2025 active perception" shows multiple papers that use RL, VLA, reasoning to do active perception on a variety of tasks on real world robots.

While at the time of writing the authors may not have known these works, it's clear the broader robot learning / RL / VLA community has recognized the importance of active perception, and have made progress there. Another missing line of work is the recent wave of learning active perception policies via imitation learning and teleoperation platforms with head eye cameras. Finally, the active inference literature proposes an objective that is very general and does handle both active perception and decision making jointly.

**Questions:**

Aside from my concerns above:

What happens if the prediction loss is not differentiable? Would RL still work on this objective?

As mentioned in the concerns, it would have been nice to have tasks where active perception is part of completing the task, but not the end goal. And comparing against a baseline that just gets task reward.

Are there videos of the learned behaviors? Is there anything interesting there, like the agent learns some efficient exploration strategy?

The experimental section gets a bit repetitive. I wish there were more qualitative analysis and visualizations instead of just comparing final accuracy scores. For example, what behaviors are learned, visualizations of the exploration behavior of the policies, etc.

---

Overall, I think this paper needs some rewriting and tweaks on the experiments, and I look forward to the changes. I like the overall direction, as the intersection of tactile sensing and active perception is under-explored, and is likely to be very important as humanoids/dexterous manipulation  become popular.

---

> ### Author Response · Authors · 2025-11-22
>
> Thank you for your detailed remarks and constructive criticism!
> In particular, we thank you for highlighting the relevance and timeliness of our paper, for noting that it is well-written, and for acknowledging that our approach is sensible.
> We would like to address your concerns and questions one by one in the following.
>
> ## On the generality of APPLE and the scope of this work
>
> We would like to emphasize that this work focuses on problems where perception is the primary objective.
> Hence, in our setting, the agent is tasked with inferring a specific property of the environment, whose ground-truth value we assume to be known at training time.
> Given this setting, we argue that the further fundamental assumptions APPLE makes are relatively mild:
> - We expect the environment to behave like a POMDP
> - We expect the (prediction-)loss function to be known to the agent and differentiable
> Note that only the loss function itself must be differentiable, and only with respect to the agent's prediction.
> Independent of task and environment, loss functions commonly used in supervised deep learning (e.g., cross-entropy loss, mean squared error, mean absolute error, etc.) usually fulfill this constraint.
> Additionally, to clarify, our method does require the agent to perceive the entire state.
> $y_t$ can be any property, e.g., the pose, class, or volume of an object, as is the case in our experiments.
>
> We fully agree that an exciting future research direction is to examine tasks in which active perception serves only as an intermediate step towards the solution.
> Although we deem an investigation of this kind of task to be outside the scope of this work, we added a small experiment to pre-validate the feasibility of using APPLE in this setting.
> In Appendix C.4, there is now an experiment, which we call CircleSquareHideAndSeek.
> In this task, the agent receives a reward for being close to squares and a penalty for being close to circles.
> The optimal strategy involves classifying which shape is present in the current episode and then staying near it if it is a square and moving away if it is a circle.
> We test two variants of our agent, one that uses just the RL-reward and one that optimizes our perception loss as well.
> We observe that only the variant of our agent with perception loss is able to solve the task, while the other one learns no meaningful behavior within 1 million time steps.
> While this experiment is too small to draw any definitive conclusions, it shows that learning tasks involving active perception by pure RL can be challenging, even for simple tasks.
> We believe that the main issue is the long-horizon dependency between acting to perceive to act, which the agent has to learn to solve tasks involving active perception.
> Hence, although fully end-to-end RL is theoretically the most general approach, in practice it often becomes intractable as the agent must simultaneously discover the relevant latent variables and learn the task policy, which frequently leads to instability and poor exploration.
> In contrast, adding structure in the form of a supervised perception loss provides an inductive bias that guides the agent towards discovering information relevant for solving the task.
>
> Finally, we would like to clarify that we do not claim that APPLE generalizes beyond our definition of active perception, nor do we claim that APPLE solves active perception in any way.
> Instead, our goal is to propose a step toward broader and less task-specific active perception methods.
> We have revised the introduction to ensure the framing is clear and not misleading.
>
>
> ## Additional baselines
>
> Thank you for your suggestion!
> We have added an experiment with a heuristic grid-search policy as a baseline for our CircleSquare experiment (see Appendix C.2, Fig. 17).
> Although this experiment is fairly simple, the baseline stagnates at around 73% accuracy, as the lower number of available steps per episode (16) is not enough to exhaustively search the environment.
> Information gain is, unfortunately, not straightforward to compute in our setting with image-based inputs, as this would require a probabilistic forward predictive model.

---

> ### Author Response · Authors · 2025-11-22
>
> ## Related works
>
> We fully acknowledge that active perception is an active area of research, with numerous papers having been published and being published today.
> With the part of the text you quoted, we do not mean to diminish prior advances in the area.
> Rather, we want to highlight that, to our knowledge, there exists no single active perception method that has been shown to work on a wide range of different sensors, environments, and tasks.
> However, to eventually deploy active perception methods in diverse, multi-modal real-world environments, having a system capable of learning active perception independently of tasks is crucial.
> With our work, we take a step in this direction by asking: _Can we learn active perception policies without making strong assumptions about sensing modalities and tasks in the algorithm design?_
> To clarify this point, we have added details to each active perception work cited in our related work section to highlight the assumptions it makes.
>
> Regarding the works from CoRL 2025 and NeurIPS 2025, we want to note that these works were not publicly available at the time of submission, and even those that were would be considered concurrent work by the [ICLR guidelines](https://iclr.cc/Conferences/2025/FAQ).
> Nevertheless, we added two works from these conferences, which we found relevant for our work [1-2], to our related works section.
>
> Furthermore, thank you for pointing out this missing line of work on imitation learning for active perception!
> We also added a few works to our related works section [2-6].
> If you are aware of additional relevant works in this area that you would like to see included, we are happy to consider them as well.
>
> Finally, regarding active inference, we are assuming you are referring to their Expected Free Energy formalism, which combines an extrinsic reward with a mutual information term to guide the agent towards informative states.
> While this formalism might seem well-suited for our objective, its implementation also comes with significant challenges.
> Crucially, to compute the mutual information of future actions, a probabilistic model of the environment is required, which, in our setting, the agent must learn from data.
> Once such a model is available, a way must be found to approximate mutual information, as its closed form is intractable for any non-linear or non-Gaussian model.
> While work in this general direction exists [7], we are not aware of any off-the-shelf methods with which to compare our method.
>
>
> ## What if the prediction loss is non-differentiable
>
> We added an experiment variant of our two-class CircleSquare task, in which we use our combined reward but interpret it as a pure RL reward, to Appendix C.2 (Fig. 16)
> Hence, we are effectively not utilizing the gradient, and the agent treats its prediction of the class label as another action variable.
> Or SAC configuration still slowly learns to solve the task but eventually becomes unstable, while our CrossQ configuration stagnates at random guessing for one million steps, underlining the importance of using the gradient of the loss function for effective learning.
>
>
> ## Additional task in which active perception is not the main goal
>
> As noted above, we have added the CircleSquareHideAndSeek task to Appendix C.4.
>
> ## Video and Qualitative analysis
>
> Thank you for pointing out the lack of qualitative analysis in the main paper.
> Due to space constraints, we had to place two figures showing the learned exploration strategies of CircleSquare and Toolbox into the appendix.
> We now reference them in the main body and have added a brief explanation of the learned strategies to each experiment.
>
> Additionally, we have included a video in the supplementary material.
> In this video, we provide a brief overview of our approach, present our experiments, and discuss some of the learned policies.
>
> Regarding learned strategies, depending on the task, we observe that the agent learns behaviors that one might consider intuitively correct, as well as behaviors that lack an obvious pattern.
> For example, in the CircleSquare task, the agent consistently learns to follow the color gradient as quickly as possible and then maintain its position above the object until the end of the episode.
> This behavior is illustrated in Figure 8.
> In the Toolbox environment, the agent learns a circular search pattern to initially find the wrench.
> Afterwards, it slides along the handle towards one of its ends to disambiguate the orientation and longitudinal position.
> We illustrate this behavior in Figure 12.
>
> In the case of the digit perception tasks (TactileMNIST and TactileMNISTVolume), the patterns in the learned strategies are less clear.
> Generally, the agents tend to move towards the center of the platforms first, as in most episodes, the objects reach into the center region.
> Once the object is found, the agent sometimes follows edges, but oftentimes the exploration seems to follow no obvious pattern anymore.

---

> ### Author Response · Authors · 2025-11-22
>
> ## References
>
> [1] Hu, Edward S., et al. "Real-World Reinforcement Learning of Active Perception Behaviors." The Thirty-ninth Annual Conference on Neural Information Processing Systems.\
> [2] Xiong, Haoyu, Xiaomeng, Xu, Jimmy, Wu, Yifan, Hou, Jeannette, Bohg, Shuran, Song. "Vision in Action: Learning Active Perception from Human Demonstrations." Proceedings of The 9th Conference on Robot Learning. PMLR, 2025.\
> [3] Liu, Shaopeng, Chao Huang, and Hailong Huang. "Behavior Cloning-Based Active Scene Recognition via Generated Expert Data With Revision and Prediction for Domestic Robots." IEEE Transactions on Robotics (2025).\
> [4] Chuang, Ian, et al. "Active vision might be all you need: Exploring active vision in bimanual robotic manipulation." 2025 IEEE International Conference on Robotics and Automation (ICRA). IEEE, 2025.\
> [5] Yang, Ning, et al. "Long-Term Active Object Detection for Service Robots: Using Generative Adversarial Imitation Learning With Contextualized Memory Graph." IEEE Transactions on Industrial Electronics (2024).\
> [6] Dai, Xu-Yang, et al. "Camera view planning based on generative adversarial imitation learning in indoor active exploration." Applied soft computing 129 (2022): 109621.\
> [7] Schneider, Tim, et al. "Active inference for robotic manipulation." arXiv preprint arXiv:2206.10313 (2022).

---

### Official Review · Reviewer_WrTJ · 2025-10-31

**Soundness:** 3
**Presentation:** 3
**Contribution:** 3
**Rating:** 8
**Confidence:** 5

**Summary:**

This paper proposes a general RL framework for task-agnostic exploration using intrinsic rewards derived from a jointly optimized prediction model. The authors instantiate an active perception setting with tactile sensing for 3D-MNIST digit classification and evaluate their methods (both SAC and CrossQ variants) on a shared Transformer architecture. Experiments indicate the approach is effective and the architectural choices are reasonable.

**Strengths:**

- A general formulation with a clear, well-motivated architecture. Clear, principled formulation of active perception as minimizing a supervised loss inside an interactive POMDP, with a clean gradient decomposition (policy gradient minus prediction-loss gradient). This is a neat unification that explains the joint training signal succinctly.
- The paper is well written with clear presentation.
- Robotic scenarios are relevant and valuable for tactile-enabled active perception with a broad set of tactile tasks (classification, regression, localization).

**Weaknesses:**

- The claim of “a novel framework that leverages RL to address a range of different active perception problems” could be more specific. Which component is novel—the unified formulation (e.g., Eq. 3), the Transformer with shared components, or something else? Clarifying this and discussing generalizability e.g. with other modalities beyond tactile sensing would help.

- Additional discussion/experiments comparing sparse (more realistic when labels arrive late in real-world settings) vs. dense rewards would strengthen claims about generality.

- Including related baselines or discussion would be helpful, e.g., RL exploration with haptics as reward [1] and earlier related work [2].

- More discussion of real-world deployment (learning time, sim-to-real if any transfer is intended) would be valuable, since real-world “exploration” can be costly or impractical.

- Minor: clarifying the definition and role of $\tilde{r}$ in Eq. 1 would aid interpretability.



References

[1] Rajeswar, Sai, Cyril Ibrahim, Nitin Surya, Florian Golemo, David Vazquez, Aaron Courville, and Pedro O. Pinheiro. "Haptics-based curiosity for sparse-reward tasks." In Conference on Robot Learning, pp. 395-405. PMLR, 2022.

[2] Li, Mengdi, Xufeng Zhao, Jae Hee Lee, Cornelius Weber, and Stefan Wermter. "Internally rewarded reinforcement learning." In International Conference on Machine Learning, pp. 20556-20574. PMLR, 2023.

**Questions:**

- Relation to IRRL: How does your RL-based active perception framework compare to the IRRL framework (Ref. [2]) in terms of objective, training signal?


- Question on the backbone choice: the tasks in this submission are with relatively small exploration spaces, is the Transformer architecture superior to an LSTM-based alternative, and why?


- Real-world deployment:
(1) Is it advisable to include warm-up strategies (e.g., demonstrations, heuristic exploration priors) for the policy, or analogous prior/knowledge injection into the prediction module?
(2) In practical deployments, access to ground-truth labels during exploration is limited or delayed. How do you handle sparse/late/partial labels, and what would the deployment workflow look like?

---

> ### Author Response · Authors · 2025-11-22
>
> Thank you for the thoughtful and detailed review!
> We appreciate your positive assessment of the paper.
> We are especially glad that you found the unified formulation of active perception within an interactive POMDP well-motivated.
> We also appreciate your recognition of the relevance of our tactile robotic scenarios and tasks evaluated.
>
> We address your questions and concerns below.
>
> ## Clarification of contribution
>
> Our contributions are now stated more explicitly in the introduction. But to better elaborate on that, we:
>
> - Propose a novel **unified** formulation of active perception as a POMDP with an objective grounded directly in minimizing supervised prediction loss.
> - To the best of our knowledge, ours is the first application of off-policy RL with a transformer-based architecture in this active perception setting.
> - Additionally, we propose as few assumptions as possible by avoiding handcrafted heuristics and supporting both vector and vision encoders for perception modalities. The method also handles dynamic scenes where the target object may move, demonstrating further robustness.
>
>
> ## Generalizability to other sensor modalities
>
> Regarding the question of generalizability beyond tactile sensing, we emphasize that our framework is designed to handle a broad range of perception modalities.
> Our perception architecture includes (i) a transformer module for generic vector observations, and (ii) an optional vision transformer that can process image-based inputs when available.
> This design makes our framework compatible with various sensor modalities commonly encountered in active perception.
>
> To further demonstrate this step towards generality, we have also added a new experiment on CIFAR-10, in which the agent must classify an image while being restricted to a small movable observation window. This setting demonstrates that the method can be applied directly to vision-based active perception without requiring architectural changes.
> The details and results are included in Appendix B.5 and C.6.
>
>
> ## On sparse rewards and delayed or partial labels
>
> We would like to emphasize that our approach does not require the labels to be available during roll-outs, as they are only used during training.
> Hence, as long as the labels become available eventually, the corresponding trajectory can be annotated with them and be stored in the replay buffer, and the training can proceed normally.
> Nevertheless, the robustness of our method to sparse rewards remains an interesting question, as we may encounter settings where labels are not available for parts of the trajectory.
> One such setting is in the case of tactile pose tracking, where the ground truth is generated by an external tracking system that may lose tracking from time to time due to occlusions.
> To support this case in our formalism, we can store a boolean variable $w_t \in \{0, 1\}$, indicating whether the object was tracked at time step $t$ ($w_t=1$) or not ($w_t=0$) alongside the perception targets $\overset{\ast}{y}_t$.
> With a tracking-aware loss function $\ell ((\overset{\ast}{y}_t, w_t), y_t) := w_t MSE(\overset{\ast}{y}_t, y_t)$, we can now make sure that only those steps get considered in the prediction loss in which the tracking was successful.
>
> To briefly evaluate the feasibility of using APPLE with sparse rewards, we conducted a small experiment on our CircleSquare environment (see Appendix C.3).
> In this experiment, we made the reward sparse by using the above loss function and setting $w_t = 0$ for all time steps except the last one, where we set $w_t = 1$.
> Essentially, the agent receives its perception reward only for the prediction it makes in the final time step.
> We observe that both APPLE variants continue to learn to solve the task, although our CrossQ variant converges slightly more slowly.

---

> ### Author Response · Authors · 2025-11-22
>
> ## Suggested baselines and relation to IRRL
>
> Thank you for pointing out these references to us!
> We have added both to our related works section.
>
> Regarding IRRL [1], it presents a framework for methods in which the agent generates the reward itself using an internal model.
> This internal model is assumed to be a discriminator, which predicts a posterior distribution over some label, and the reward is computed as the mutual information between the trajectory and the label under this distribution.
> Our approach is similar in the sense that the reward our agent receives is also largely dependent on the output of its (internal) prediction model.
> However, fundamentally, APPLE is optimizing arbitrary loss functions, while IRRL is defined for maximizing mutual information.
> Interestingly, in the case where our loss function is a cross-entropy loss, IRRL and APPLE optimize nearly the same objective function, assuming a uniform prior distribution over labels and that APPLE uses a constant zero RL reward.
>
> To stabilize their training, IRRL applies two tricks: dropping the log operators of their probability terms in their objective and clipping the resulting value at zero.
> In APPLE, we did not observe the instabilities they are describing, which made such tricks unnecessary for us.
> One explanation might be that we are using off-policy RL, while IRRL seems to demonstrate on-policy RL methods only, which are known to require tricks like clipping in their objective function [2].
>
> Hence, in summary, IRRL provides a common umbrella for on-policy active perception RL methods, such as RAM [3] (which our baseline HAM is based on), while improving their convergence through the aforementioned tricks.
>
>
> ## Discussion of real-world deployment
>
> While outside the scope of this work, we plan to investigate the real-world deployment of APPLE in the future.
> As with many RL-based approaches, the sample efficiency of APPLE currently makes direct deployment on a real system infeasible.
> However, with the emergence of more realistic tactile simulators [4], sim-to-real transfer of learned tactile manipulation policies might soon become more feasible.
> At the same time, other strategies may be employed to enhance APPLE's sample efficiency in the real world.
> Aside from the policy warm-up strategies you already mentioned, using pre-trained encoders might also improve both robustness and sample efficiency.
> We have added a paragraph in the conclusion discussing the future prospects of real-world deployment.
>
> Regarding the handling of sparse/delayed/partial labels, see our comment above.
>
>
> ## Clarify the definition and roll of $\tilde{r}$ in Eq. 1
>
> $\tilde{r}$ is the overall reward function that the agent is trying to maximize.
> It is defined as $\tilde{r} (h_t, \overset{\ast}{y}_t, a_t, y_t) = r(h_t, a_t) - \ell(\overset{\ast}{y}_t, y_t)$ and explained in the paragraph above Eq. 1.
> We slightly modified this paragraph to make the role of $\tilde{r}$ clearer.
> Is there anything particular about the definition and explanation of $\tilde{r}$ you find confusing?
> If yes, we are happy to update the paragraph for more clarity.
>
>
> ## Transformer vs LSTM
>
> Throughout this project, we have experimented with LSTMs instead of transformers, as they offer significant advantages w.r.t computational efficiency, especially for longer horizons.
> However, despite extensive hyperparameter search, we did not achieve compelling performance on our CircleSquare benchmark, which is the easiest of our tasks aside from the MHSB task.
> Hence, we decided to use a transformer as a backend.
> In addition to the experiment in Figure 14, where we compare PPO with an LSTM model and PPO with a transformer model on the MHSB task, we have now added an experiment with APPLE using an LSTM on the CircleSquare task to Appendix C.2 (See Fig. 15)
>
>
> ## References
> [1] Li, Mengdi, et al. "Internally rewarded reinforcement learning." International Conference on Machine Learning. PMLR, 2023. \
> [2] Schulman, John, et al. "Proximal policy optimization algorithms." arXiv preprint arXiv:1707.06347 (2017). \
> [3] Mnih, Volodymyr, et al. "Recurrent models of visual attention." Advances in neural information processing systems 27 (2014). \
> [4] Nguyen, Duc Huy, et al. "TacEx: GelSight Tactile Simulation in Isaac Sim--Combining Soft-Body and Visuotactile Simulators." arXiv preprint arXiv:2411.04776 (2024).

---

### Official Review · Reviewer_bEC3 · 2025-11-05

**Soundness:** 3
**Presentation:** 2
**Contribution:** 3
**Rating:** 6
**Confidence:** 4

**Summary:**

This paper introduces APPLE, a reinforcement learning framework designed for active perception tasks. The core formulation treats active perception as a POMDP and jointly optimizes a transformer-based policy for action selection and a perception module for task prediction, using a reward signal that incorporates task-specific differentiable losses. The method is evaluated on multiple simulated benchmarks covering classification, regression, and pose estimation tasks. Results indicate that APPLE outperforms both a random baseline and a prior method (HAM), demonstrating its ability to learn effective exploration strategies without relying on task-specific heuristics.

**Strengths:**

The principal strength lies in the consistent performance across multiple tasks without requiring task-specific modifications, demonstrating good innovation.
Extensive experiments across five benchmarks offer convincing evidence of the framework's capability and robustness.
The comparison with HAM effectively underscores the benefits of off-policy methods in improving sample efficiency.

**Weaknesses:**

The comparison is largely limited to a random baseline and HAM. Evaluation against other non-RL active perception methods would strengthen the validity of the claims.
The paper lacks thorough ablation analysis. For instance, the importance of the transformer architecture remains unclear.

**Questions:**

The "RL reward" is mentioned but not specified in detail for each task. How was this term defined across the different benchmarks?
The transformer backbone introduces considerable complexity. Was a comparison performed with a simpler recurrent model to ensure that performance gains stem from the proposed framework rather than the representational capacity of the transformer?

---

> ### Author Response · Authors · 2025-11-22
>
> Thank you for acknowledging the strengths of our work and providing constructive criticism.
> Particularly, we appreciate your recognition that APPLE performs consistently across diverse tasks without task-specific tuning, and that this demonstrates good innovation.
> We will address each of your points in the following:
>
> ## Providing additional baselines
>
> Regarding the request for more baselines, we agree that comparing against additional active perception methods could further strengthen the evaluation.
> However, to the best of our knowledge, there are currently no further off-the-shelf active perception algorithms capable of handling our setting (vector input + vision observations, dynamic scenes where objects may move, and capable of handling classification and regression).
> Nevertheless, to address the concerns, we have added some additional baselines on the Circle–Square task:
>
> - Fixed trajectory policy: In this one, we have defined a fixed grid search pattern.
>   Here, the policy will no longer decide on the actions, but instead, the agent will follow a predetermined grid-search trajectory, while the perception module is trained exactly as in APPLE (see Appendix C.2, Fig. 17).
> - Pure RL baseline: This one is a variant of our method trained with CrossQ.
>   What makes it different from APPLE is that we do not give the agent information about the loss (see Appendix C.2, Fig. 16).
>
>
> ## Importance of transformer architecture
> To investigate the role of the transformer backbone, we have added additional experiments on our CircleSquare task, in which we test our method with an LSTM backbone (see Appendix (See Appendix C.2, Fig. 15).
> Furthermore, we refer to the PPO results in Appendix (Appendix C.2, Fig. 14), where we compared PPO on CircleSquare with a transformer backbone to an LSTM backbone.
> We observe that the LSTM backbone is consistently outperformed by our transformer backbone, despite extensive hyperparameter tuning.
> Since we did not achieve good performance with LSTMs on our easiest task, we decided not to pursue the option of using LSTMs further.
>
> ## What is the RL reward in our tasks?
> In Appendix B, we provide more details on the environments we tested APPLE on, including the RL reward used.
> Specifically, we use the RL reward only for regularizing the action, that is $r(h_t, a_t)=10^{-3} \lVert a_t \rVert^2$.

---

### Author Response · Authors · 2025-11-22

We thank all reviewers for their constructive feedback and detailed comments!
We are encouraged that multiple reviewers highlighted the principled unification of prediction and control (WrTJ, yWQL), the strength and diversity of tasks (bEC3, WrTJ), the clarity of our formulation (WrTJ, yWQL), and the promise of active perception in tactile domains (LSx3).
We also appreciate the positive remarks on the writing, reproducibility, architectural choices, and relevance to future robotic systems.

Below, we address some overall concerns:

## Scope: Active Perception
We would like to clarify that in this work, we are solely concerned with the problem of _Active Perception_, which refers to the process of acting to extract desired information from the environment.
While many tasks involve active perception as an intermediate step, this work specifically examines problems where perception is the primary objective, such as classifying a given object or estimating the pose of an object from tactile data.
This limited scope enables us to examine active perception in isolation, without entanglement with other task-performance factors.

Furthermore, although the fields of active perception and intrinsic motivation-driven reinforcement learning are adjacent, we would like to emphasize that APPLE does not fall under the umbrella of intrinsic motivation methods.
Intrinsic motivation methods are typically concerned with accelerating RL training by encouraging the agent to explore novel states, typically in MDP settings.
In our case, the agent uses exploration solely to reduce uncertainty about a target property within a POMDP, thereby maximizing its immediate returns in the current episode.
The two concepts are fully orthogonal and could even be combined by adding an intrinsic reward term to APPLE's reward function, potentially speeding up its learning.

## Generality of the method

We also clarify that we are not claiming that our method is a general solution to active perception.
Our long-term vision for this line of work is to eventually develop agents capable of active perception across a wide range of domains and sensing modalities, including tactile sensing, vision, sound, and possibly even smell.
To make progress toward this goal, we designed APPLE with generality in mind, making as few assumptions as possible about the problem within our definition of active perception and in the context of interactive supervised learning.
APPLE is by design not bound to a specific problem or sensor type and is formally compatible with any visual or vector-type data, which covers a broad range of commonly used sensors.
Yet, we do not claim that it is compatible with every sensor, nor do we claim that APPLE is capable of learning with any sensor it is compatible with.
We argue, however, that our work is a step towards more general active perception methods not tied to a specific problem or sensor type.
As methods in this domain are usually defined for specific tasks, our contribution consists of defining a framework general enough to support a variety of sensors and tasks.
This includes multiple classification and regression tasks, and both vision and tactile tasks, both with sensor arrays or higher-dimensional images.
To avoid any misunderstanding, we have clarified our claims in the Introduction and Limitations section (Appendix D).
We are happy to make further changes, should there be other parts in the paper where the scope and claims of this work may be misunderstood.

## Summary of changes

For an overview of the changes, we also summarise here our modifications to the paper:
- Introduction: We **clarified the list of contributions, assumptions, and claims about generality**.
- Related works: We **extended the discussion about previous works**, including additional works on RL for active perception, works from CoRL 2025 and NeurIPS 2025, and IRRL.
- Problem Statement: We clarified $\tilde{r}$
- Discussion: We **added a brief qualitative discussion** on the emerging behaviour of our policies and referenced the existing qualitative insights in the appendix.
- We evaluated our method in a **new environment for natural image classification** (the CIFAR-10 classification task, see Apdx. B.5, C.6)
- We added an **ablation study of the CircleSquare environment** (Apdx. C.2): We have evaluated APPLE with new ablations and baselines, including a study on the importance of the transformer backbone, a baseline with pure RL, and comparison with a fixed trajectory policy.
- Evaluation in a **version of the CircleSquare environment with sparse rewards** (Apdx. C.3)
- Additional evaluation on APPLE's ability to solve a downstream task more efficiently and **introduction of the CircleSquareHideAndSeek** task (Apdx. C.4).
- We clarified our limitations (Apdx. D).
- **Discussion on the connection between APPLE and curiosity-based intrinsic reward methods** (Apdx. H).
- **We added a video** showcasing the method and learned behavior

---

### Author Response · Authors · 2025-11-29
**Author Rebuttal Summary (Part 2)**

- _Reviewers bEC3 and LSx3 requested additional baselines._
  We implemented an additional heuristic grid-search baseline as suggested by Reviewer LSx3, and we also now compare our approach against pure-RL baselines, which do not utilize the target label.
  Both baselines further underscore the efficacy of our method.

- _Reviewers bEC3 and WrTJ requested an ablation of the transformer architecture_, which we, among other ablations, now provide in Apdx. C.2.

- _Reviewer LSx3 requested a qualitative study of learned behaviors_, which we added to our experiments section and the appendix.
  Furthermore, we have added a video that explains our method and showcases some of the learned behaviors.

- _Reviewer WrTJ requested that we clarify our contributions_, which we did.

- _Reviewer WrTJ requested an experiment evaluating our method in a sparse-reward setting_, which we now provide in Apdx. C.3.

- _Reviewer WrTJ suggested additional related work_ [1,2], which we added to the paper.
  We dedicated one paragraph to IRRL [2], which we deemed particularly relevant to our work.

- _Reviewer WrTJ requested a discussion of real-world deployment_, which we have now added to our conclusion.

We believe that throughout this rebuttal process, we have thoroughly addressed most of the reviewers' concerns and substantially improved the manuscript.
Therefore, we would like to thank the reviewers sincerely once again for their time and effort.
Unfortunately, for reasons beyond everyone's control, we were unable to engage in discussion with the reviewers and convince them to raise their scores.
Nevertheless, we hope to convince you that substantial progress has been made on this paper, and we would appreciate your consideration of this in your final decision.

Best,
The authors

### References

[1] Rajeswar, Sai, Cyril Ibrahim, Nitin Surya, Florian Golemo, David Vazquez, Aaron Courville, and Pedro O. Pinheiro. "Haptics-based curiosity for sparse-reward tasks." In Conference on Robot Learning, pp. 395-405. PMLR, 2022.\
[2] Li, Mengdi, Xufeng Zhao, Jae Hee Lee, Cornelius Weber, and Stefan Wermter. "Internally rewarded reinforcement learning." In International Conference on Machine Learning, pp. 20556-20574. PMLR, 2023.

---

### Author Response · Authors · 2025-11-29
**Author Rebuttal Summary (Part 1)**

Dear AC,

Since the reviewer discussion was cancelled due to the OpenReview data leak, and the responsibility of considering our rebuttal now fully lies with you, we would like to provide a brief summary of the weaknesses highlighted by the reviewers and how we addressed them.
For more detailed and comprehensive explanations, we refer to our individual responses.

- _Reviewers LSx3 and yWQL found parts of the framing of our method w.r.t its generality misleading._
  We highlighted that we do not claim that our method works on arbitrary active perception problems.
  Rather, we view this work as a step _towards_ more general active perception, as within our defined active perception setting, we strive to make as few assumptions as possible about the sensors and tasks.
  **We do not claim to have achieved general active perception capabilities.**
  Nevertheless, we agree that some statements in our initial submission may be perceived as misleading, and we have revised those accordingly to prevent any further misunderstanding.

- _Reviewer LSx3 was unsatisfied with our related work section and suggested adding works from CoRL/NeurIPS 2025, as well as works on imitation learning for active perception._
  We informed the reviewer that works from these conferences were not publicly available at the time of submission and would therefore fall under concurrent work according to the ICLR guidelines; however, we added them to the revised manuscript nonetheless.
  Additionally, we expanded our related works section to include works on imitation learning for active perception.

- _Reviewer yWQL claimed that APPLE lacks novelty._
  However, aside from stating that research in active perception has been conducted in the past (a fact with which we fully agree, see our related works section), no concrete evidence is provided to back this claim.
  We asked the reviewer to provide references to works close enough to our work to raise concerns about novelty, but received no reply before reviewer comments were disabled.
  In particular, we are concerned by the fact that in this review, our method is seemingly conflated with curiosity-based intrinsic reward methods such as ICM and RND.
  As we explain in our general response, our work is fully orthogonal to ICM and RND, as the objectives and settings are fundamentally different: we explore within episodes of POMDPs to infer task-relevant properties, while ICM and RND explore across episodes of MDPs to learn policies.
  Since we understand that, on first glance, the reward function of our method might resemble those used in intrinsic-reward RL methods, we have clarified the connection between those and our method in Apdx. D.
  Furthermore, we have clarified our list of contributions in the manuscript.

- _Reviewer yWQL criticized the scope of our experimental evaluation, claiming we would test our method only in simulation and only on a single object._
  We fully acknowledge that we did not test our method in the real world, which we already clearly stated in our conclusion and limitations section.
  However, we also highlight that our contributions are of an algorithmic nature, and thus, we deem robotic evaluation to be outside the scope of this work.
  Furthermore, the TactileMNIST benchmark we test our method on contains more than **13,000** distinct objects, with different training and validation splits.
  We also evaluate on the MHSB task, which involves four different shapes.
  The only task involving a single object is the Toolbox task, which does indeed include only one wrench.
  We believe the reviewer may have interpreted this as representative of all tasks, whereas APPLE is evaluated on and demonstrates robustness across diverse object shapes, even if this is not explicitly illustrated in the Toolbox case alone.

- _Reviewer yWQL stated our paper "emphasized multi-sensor and task-agnostic capabilities" without providing corresponding experiments._
  **We do not claim to have multi-sensor and task-agnostic capabilities.**
  We state that our approach is compatible with different sensor types, which we back by experiments with different (single) sensors (pixel glimpse, GelSight Mini, the Myrmex taxel array, and proprioception).
  Furthermore, although APPLE tries to make as few assumptions about the task as possible, our experiments, which indeed cover a range of tasks (shape classification, volume estimation, and pose estimation), are not comprehensive enough to support any claims of task-agnostic capabilities.
  For that reason, we never claim APPLE has task-agnostic capabilities.

- _Reviewer yWQL asked if APPLE could handle sensor noise._
  In response, we added an experiment where the agent views a movable glimpse of a CIFAR-10 image and has to classify it.
  Due to the nature of natural images, this task demonstrates that APPLE is capable of handling some degree of sensor noise.

---

### Meta-Review · Area_Chair_4m3o · 2026-01-05

**Summary:**

My recommendation for the paper is Accept.

This paper initially received mixed scores (2, 4, 6, 8), mainly due to the concern on the misleading writing in the original draft that confused reviewers about the problem setup and contributions of this paper. The rebuttal response was successful in resolving these concerns -- it properly set & narrowed the scope of the paper, clarified the contributions, and provided additional experiments to support the contributions.

After reading the reviews, rebuttal response, and the paper, it was still difficult to finalize the recommendation for this paper because of the preliminary nature of experimental results in the paper. However, I ended up recommending the acceptance of this paper because I think the ICLR community can benefit from having a further discussion on the area of active vision based on the general framework proposed in this paper. I recommend the authors to further clarify the scope and contributions of the paper for the camera-ready version.

**Reviewer Concerns:**

Concerns resolved by the rebuttal response
- Additional baselines
- Ablation study on transformer architecture
- Clarification on novelties
- Additional discussion on related work
- Discussion on real-world evaluation
- Misleading / overstated claim on the generality of the method
- Lack of novelty

Outstanding concerns
- Missing real-world evaluation: Authors did raise a point that this is out of scope at the current status, which is a fair point, but anyways this was not resolved.

**Reviewer Scores:**

- Reviewer bEC3 (score 6): I expect the reviewer to maintain their positive score
- Reviewer WrTJ (score 8): I expect the reviewer to maintain their positive score
- Reviewer LSx3 (score 4): I expect the reviewer to increase their score, as the reviewer mostly requested rewriting the paper and including further discussion on related work
- Reviewer yWQL (score 2): I expect the reviewer to maintain their negative score, as real-world experiments are not provided.

---

### Decision · Program_Chairs · 2026-01-26

Accept (Poster)